# ADAPTS: Automated deconvolution augmentation of profiles for tissue specific cells

**Samuel A. Danziger**[1]*, **David L. Gibbs**[2], **Ilya Shmulevich**[2], **Mark McConnell**[1], **Matthew W. B. Trotter**[1,3], **Frank Schmitz**[1], **David J. Reiss**[1], **Alexander V. Ratushny**[1]*

**1** Celgene Corporation, Seattle, Washington, United States of America, **2** Institute for Systems Biology, Seattle, Washington, United States of America, **3** Celgene Institute for Translational Research Europe, Seville, Sevilla, Spain

* sdanziger@celgene.com (SAD); aratushny@celgene.com (AVR)

**Data Availability Statement:** The data presented in the publication has been made available on GitHub as R data packages. This makes it easy to replicate the results presented in the vignettes.

## Abstract

Immune cell infiltration of tumors and the tumor microenvironment can be an important component for determining patient outcomes. For example, immune and stromal cell presence inferred by deconvolving patient gene expression data may help identify high risk patients or suggest a course of treatment. One particularly powerful family of deconvolution techniques uses signature matrices of genes that uniquely identify each cell type as determined from single cell type purified gene expression data. Many methods from this family have been recently published, often including new signature matrices appropriate for a single purpose, such as investigating a specific type of tumor. The package **ADAPTS** helps users make the most of this expanding knowledge base by introducing a framework for cell type deconvolution. **ADAPTS** implements modular tools for customizing signature matrices for new tissue types by adding custom cell types or building new matrices *de novo*, including from single cell RNAseq data. It includes a common interface to several popular deconvolution algorithms that use a signature matrix to estimate the proportion of cell types present in heterogenous samples. **ADAPTS** also implements a novel method for clustering cell types into groups that are difficult to distinguish by deconvolution and then re-splitting those clusters using hierarchical deconvolution. We demonstrate that the techniques implemented in **ADAPTS** improve the ability to reconstruct the cell types present in a single cell RNAseq data set in a blind predictive analysis. **ADAPTS** is currently available for use in **R** on CRAN and GitHub.

## Introduction

Determining cell type enrichment from gene expression data is a useful step towards determining tumor immune context [1, 2]. One family of techniques for doing this involves regression with a signature matrix, where each column represents a cell type and each row contains the average gene expression for each cell types [3, 4]. These signature matrices are constructed

https://github.com/sdanzige/ADAPTSdata https://github.com/sdanzige/ADAPTSdata2 https://github.com/sdanzige/ADAPTSdata3.

**Funding:** This work was supported by Institute for Systems Biology. The funder provided support in the form of salaries for authors DLG and IS, but did not have any additional role in the study design, data collection and analysis, decision to publish, or preparation of the manuscript. The specific roles of these authors are articulated in the 'author contributions' section. This work was supported by Celgene Corporation. The funder provided support in the form of salaries for authors SAD, MM, MWBT, FS, DJR and AVR, but did not have any additional role in the study design, data collection and analysis, or preparation of the manuscript. The specific roles of these authors are articulated in the 'author contributions' section. Celgene Corporation approved the authors' decision to publish after an internal intellectual property review.

**Competing interests:** Our commercial affiliation with the Institute for Systems Biology and Celgene Corporation has no other relevant declarations relating to employment, consultancy, patents, products in development, or marketed products. No author has any competing interest that interferes with, or could reasonably be perceived as interfering with, the full and objective presentation, peer review, editorial decision-making, or publication of research or non-research articles submitted to this journal. This (commercial affiliation) does not alter our adherence to PLOS ONE policies on sharing data and materials.

using gene expression from samples of a purified cell type. Generally, the publicly available versions of these gene expression signature matrices use immune cells purified from peripheral blood. Genes are included in these matrices based on how well they distinguish the constituent cell types. Although examples exist of both general purpose immune signature matrices, e.g. LM22 [5] and Immunostates [6], and more tissue specific ones e.g. M17 [7], these publicly available matrices are most likely not appropriate for all diseases and tissue types. One such example would be multiple myeloma whole bone marrow samples, which pose multiple challenges: both tumor and immune cells are present, immune cells may have different states than in peripheral blood, and non-immune stromal cells such as osteoblasts and adipocytes are expected play an important role in patient outcomes [8].

One straightforward solution to this problem would be to augment a signature matrix by adding cell types without adding any additional genes. For example, one might find purified adipocyte samples in a public gene expression repository and add the average expression for each gene in the matrix to create an adipocyte augmented signature matrix. While this might work, one might reasonably expect adipocytes (for example) to best be identified by genes that are different from those that best characterize leukocytes. Furthermore, it will be unclear which deconvolution algorithm would be most appropriate for applying this new signature matrix to samples. Once cell types have been deconvolved, it will also be unclear which cell types are likely to be confused due to a common lineage or other factors and how to best resolve this confusion. These problems are exacerbated by newly available single cell RNAseq data, which promises to identify the cell types that are present in a particular sample and gene expression for those cell types, but is hampered by clustering techniques that may incorrectly identify groups of cells as distinct cell types.

We have developed the **R** package **ADAPTS** (Automated Deconvolution Augmentation of Profiles for Tissue Specific cells) to help solve these problems as shown in Fig 1. **ADAPTS** is currently available on CRAN (https://cran.r-project.org/web/packages/ADAPTS) and GitHub (https://github.com/sdanzige/ADAPTS). As the package vignettes already provide step-by-step instructions for applying **ADAPTS** to the aforementioned problems, this manuscript is intended to complement the package by providing a theoretical understanding of the **ADAPTS** methodology.

## Materials and methods

**ADAPTS** aids deconvolution techniques that use a signature matrix, here denoted as $S$, where each column represents a cell type and each row contains the average gene expression in that cell type [3, 4]. These signature matrices are constructed using gene expression from samples of purified cell types, $P$, and include genes that are good for identifying cells of type $c$ where $c \in C$ and $C$ is a population of cell types to look for in a sample.

Deconvolution estimates the relative frequency of cell types in a matrix of new samples $X$ where each column is a sample and each row is a gene expression measurement according to Eq 1.

$$E = D(S, X) \tag{1}$$

Eq 1 results in a cell type estimate matrix $E$, where each column is a sample corresponding to a column in $X$, and each row is a cell type corresponding to a column in $S$ (potentially with an extra row representing an 'other' cell type not in $S$).

One straightforward method to augment a signature matrix, $S$, would be to add new cell types, $NC$, without adding any additional genes. For example, one might start with LM22 as an initial signature matrix, $S^0$, with $|g_{S^0}| = 547$ genes (rows) and $|C = 22|$ cell types (columns) and

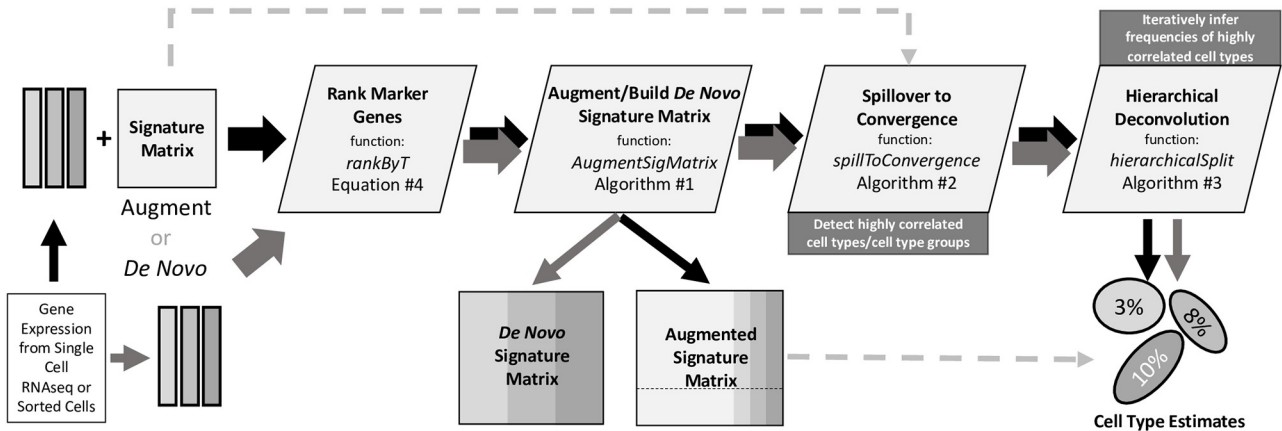

# ADAPTS Overview

**Fig 1. Overview of ADAPTS modules.** New gene expression data from cell types (e.g. from a tumor microenvironment) will be used to construct a new signature matrix *de novo* or by augmenting an existing signature matrix. First **ADAPTS** will rank marker genes for the cell types using the function *rankByT* described in Eq 4. Then **ADAPTS** adds marker genes in rank order using the function *AugmentSigMatrix* as described in Algorithm 1 resulting in a new signature matrix. This matrix may be tested for spillover between cell types using the function *spillToConvergence* described in Algorithm 2. Finally, **ADAPTS** separates cell types with heavy spillover using the hierarchical deconvolution function *hierarchicalSplit* described in Algorithm 3 to estimate the percentage of cell types present in bulk gene expression data.

augment with $c \in NC$ purified cell types. Let $NC_1$ = adipocytes and $P^1$ be an adipocyte samples matrix with $|G| = 20,000$ genes (rows) and $|J_1| = 9$ samples (columns) taken from a public gene expression repository such as ArrayExpress [9] or the Gene Expression Omnibus [10]. A new column could be constructed from $P^1$ using the average expression for each of the 547 genes $(g_1 \ldots g_{547})$ in $G_{S^0}$. Extended to all $c \in NC$, this would produce Eq 2.

$$S^1 = \begin{pmatrix} S^0_{g_1,1} & \cdots & S^0_{g_1,22} & \frac{1}{|J_1|}\sum_{j \in J_1} P^1_{g_1,j} & \cdots & \frac{1}{|J_{|NC|}|}\sum_{j \in J_{|NC|}} P^{|NC|}_{g_1,j} \\ \vdots & \ddots & \vdots & \vdots & \ddots & \vdots \\ S^0_{g_{547},1} & \cdots & S^0_{g_{547},22} & \frac{1}{|J_1|}\sum_{j \in J_1} P^1_{g_{547},j} & \cdots & \frac{1}{|J_{|NC|}|}\sum_{j \in J_{|NC|}} P^{|NC|}_{g_{547},j} \end{pmatrix} \quad (2)$$

Thus $S^1$ is a signature matrix augmented with the cell types in $NC$. While this might work, one might reasonably expect adipocytes to best be identified by genes that are different from those that best characterize the 22 cell types in $S^0$.

## Signature matrix augmentation

**ADAPTS** provides functionality for augmenting an existing cell type signature matrix with additional genes or even constructing a new signature matrix *de novo*. In addition to $S^0$ and $P^1$, this requires $S^{E0}$, an extended version $S^0$ with all genes. From this data, **ADAPTS** selected $N$

additional genes $g_{n_1} \ldots N$ to augment the signature matrix as shown in Eq 3.

$$S^i = \left( \begin{array}{ccc|ccc} S^0_{g_1,1} & \cdots & S^0_{g_1,22} & \frac{1}{|J_1|}\sum_{j \in J_1} P^1_{g_1,j} & \cdots & \frac{1}{|J_{|NC|}|}\sum_{j \in J_{|NC|}} P^{|NC|}_{g_1,j} \\ \vdots & \ddots & \vdots & \vdots & \ddots & \vdots \\ S^0_{g_{547},1} & \cdots & S^0_{g_{547},22} & \frac{1}{|J_1|}\sum_{j \in J_1} P^1_{g_{547},j} & \cdots & \frac{1}{|J_{|NC|}|}\sum_{j \in J_{|NC|}} P^{|NC|}_{g_{547},j} \\ \hline S^{E0}_{g_{n_1},1} & \cdots & S^{E0}_{g_{n_1},22} & \frac{1}{|J_1|}\sum_{j \in J_1} P^1_{g_{n_1},j} & \cdots & \frac{1}{|J_{|NC|}|}\sum_{j \in J_{|NC|}} P^{|NC|}_{g_{n_1},j} \\ \vdots & \ddots & \vdots & \vdots & \ddots & \vdots \\ S^{E0}_{g_{n_N},1} & \cdots & S^{E0}_{g_{n_N},22} & \frac{1}{|J_1|}\sum_{j \in J_1} P^1_{g_{n_N},j} & \cdots & \frac{1}{|J_{|NC|}|}\sum_{j \in J_{|NC|}} P^{|C|}_{g_{n_N},j} \end{array} \right) \quad (3)$$

**ADAPTS** helps a user construct new signature matrices with modular **R** functions and default parameters to:

1. Identify and rank significantly different genes for each cell type.

2. Evaluate the stability (condition number, $\kappa(S^x)$) of many signature matrices $S^x \in S$.

3. Smooth and normalize to meet tolerances for a robust signature matrix.

These components are combined into a single function that produces a new deconvolution matrix. First the algorithm ranks each the genes that best differentiate each cell types such that there is a ranked set of genes $g^c$ for each $c \in C$ where $C$ includes the cell types in the original signature matrix, $S^0$ as well as the new cell types $NC$. Genes, $g^c$ (where $g^c \subseteq G$ and $G$ is the set of all genes), are ranked in descending order according to scores calculated by Eq 4 and exclude any that do not pass a t-test determined false discover rate cutoff (by default, 0.3).

$$score(g_n) = \left\| log_2 \left( \frac{\frac{1}{|J_c|}\sum_{j \in J_c} P^c_{g_n j}}{\frac{1}{|J_{C-\{c\}}|}\sum_{j \in J_{C-\{c\}}} P^{C-\{c\}}_{g_n j}} \right) \right\| \quad (4)$$

Thus $g^c = sort(\forall n \in N : score(g_n^c))$ and the function $pop(g^c)$ will return and remove the gene with the largest absolute average log expression ratio between the cell type, $c$, and all other cell types, $C - \{c\}$. As shown in Algorithm 1, the matrix augmentation algorithm iteratively adds the top gene that is not already in the signature matrix from each $c \in C$ and calculates the condition number for that matrix. The augmented signature matrix is then chosen that minimizes the condition number, $CN$.

**Algorithm 1** Augment signature matrix

```
Require: S⁰, S^E0, and P {as defined for Eqs 2 and 3}
  S¹ = (S⁰|A(P_{g∈G_{s0}})) {S¹ is augmented as shown in Eq 2}
  minCN = CN₁ = κ(S¹)
  bestIndex = 1
  for i = 2: nIter do
    g_{1...N} = ∀c ∈ C: pop(g^c) {i.e. take the top gene for each cell type}
    Sⁱ = ((S^{i-1})ᵀ|A(P_{g_{1...N}})ᵀ)ᵀ {Sⁱ is augmented as shown in Eq 3}
    CNᵢ = κ(Sⁱ)
    if CNᵢ < minCN then
      minCN = CNᵢ
```

```
        bestIndex = i
      end if
    end for
    {bestIndex is recalculated after smoothing CN and optionally apply-
ing a tolerance}
    return S^bestIndex
```

In Algorithm 1: $nIter = 100$ by default, and $\kappa(s)$ returns the condition number. $A(P)$ returns the mean expression for each gene in each cell type, producing a matrix such as is shown on the right side of Eq 2. If $P$ has $|C|$ cell types, $|G|$ genes, and each cell type has some number of samples, $|P^i|$ where $i = 1: |C|$, then $A(P)$ would result in a matrix with $|C|$ columns and $|G|$ rows. When $A(P)$ is called on a matrix with one cell type, $P^i$, then $A(P^i)$ results in a matrix with one column and $|G|$.

Fig 2 shows a plot of condition numbers when adding 5 cell types to a 22 cell type signature matrix with smoothing and a 1% tolerance.

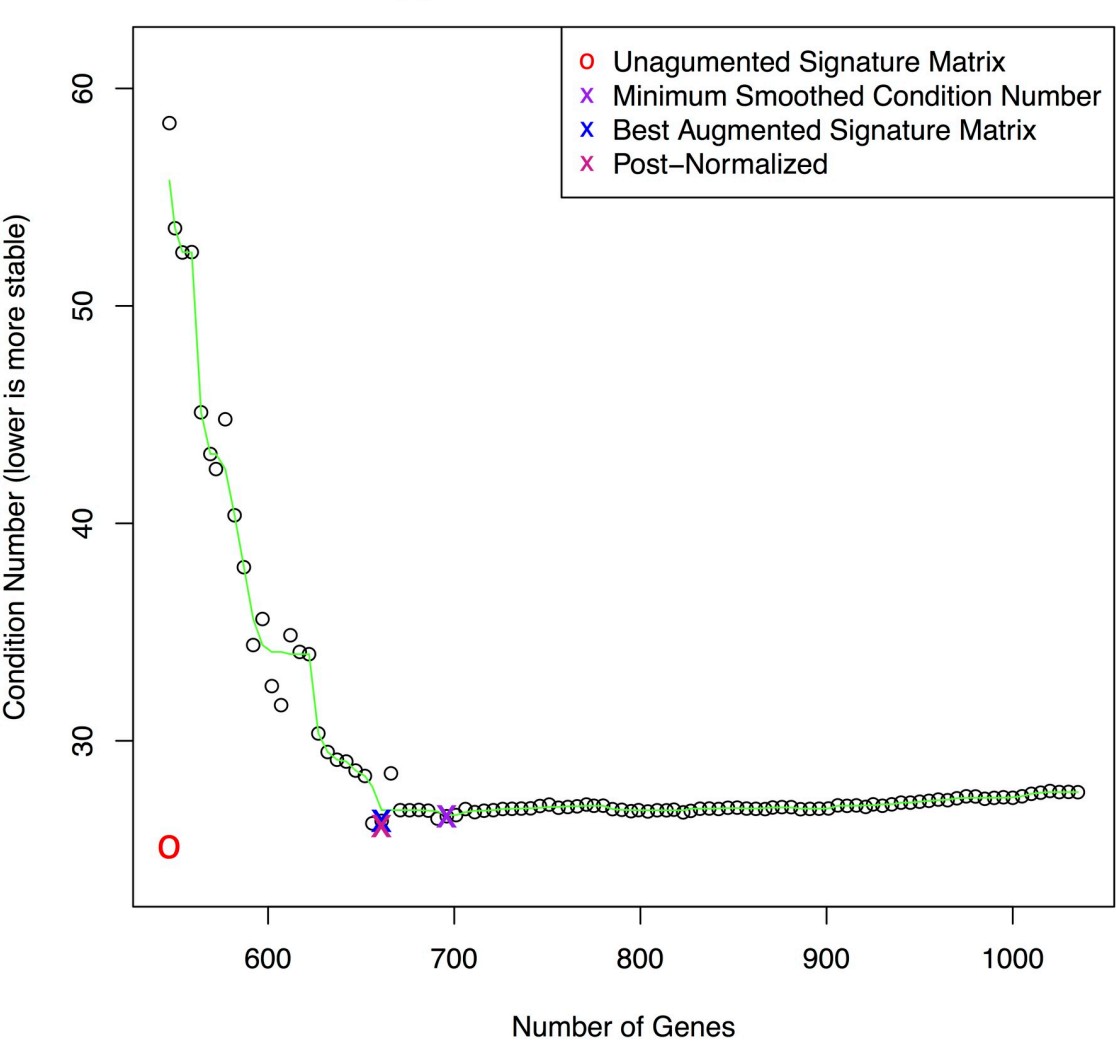

**Fig 2. MGSM27 construction.** Curve showing the selection of an optimal condition number for MGSM27.

Similarly, **ADAPTS** can be used to construct a *de novo* matrix from first principals rather than starting with a pre-calculated $S^0$. One technique is to build $S^0$ out of the $n$ (*e.g.* 100) genes that vary the most between cell types and use **ADAPTS** to augment that seed matrix. The $n$ initial genes can then be removed from the resulting signature matrix and that new signature matrix can be re-augmented by **ADAPTS**.

## Condition number minimization and smoothing

The condition number (*CN*) is calculated by the $\kappa()$ function. In linear regression, *CN* is a metric that increases with multicollinearity; in this case, how well can the signature of cell types be linearly predicted from the other cell types in the signature matrix. To illustrate this, it is helpful restate Eq 1 using a signature matrix $S$ that has the same number of genes as the data to deconvolve $X$ and use the trivial deconvolution function: $D(S, X) = S^{-1} X$. This recasts Eq 1 deconvolution as the matrix decomposition problem presented in Eq 5 [11].

$$X \approx SE \tag{5}$$

When the problem is stated in this manner, the *CN* approximately bounds the inaccuracy *E*, the estimate of cellular composition [12], and it remains meaningful if the system becomes underdetermined [13].

By definition, the condition number will increase as the system become more multicollinear. If a signature matrix is augmented with new cell types that express the genes in the matrix in a pattern similar to other cell types in the matrix, then the condition number would be expected to dramatically increase compared to the un-augmented matrix. This indicates a signature matrix lacking genes informative for differentiating multicollinear gene signatures. As Algorithm 1 iteratively adds the top gene for each cell type, the condition number would be expected to decrease as the new genes are selected to differentiate that cell type from all other cell types.

In practice, Algorithm 1 sometimes results in clearly unstable minima, where the *CN* decreases dramatically for one iteration only to increase dramatically the next. To avoid this instability, **ADAPTS** smooths the *CN* curve using Tukey's Running Median Smoothing (3RS3R) [14]. Often, the *CN*s will decrease in very small increments for many iterations before beginning to rise, resulting thousands of genes in the signature matrix. A signature matrix ($S$) with more genes ($|g_S|$) than samples in the training data ($\Sigma_{j \in J}|J_j|$) essentially represents the solution to an underdetermined system that is likely to be overfit to the training data, resulting in reduced deconvolution accuracy on new samples. To mitigate this, an optional tolerance may be applied to find the minimum number of genes that has a *CN* within some % of the true minimum. By default, **ADAPTS** uses a 1% tolerance.

## Deconvolution framework

The **ADAPTS** package includes functionality to call several different deconvolution methods using a common interface, thereby allowing a user to test new signature matrices with multiple algorithms. These function calls fit the form $D(S, X)$ presented in Eq 1.

The algorithms include:

1. **DCQ** [15]: An elastic net based deconvolution algorithm that consistently best identifies cell proportions.

2. **SVMDECON** [5]: A support vector machine based deconvolution algorithm.

3. **DeconRNASeq** [16]: A non-negative decomposition based deconvolution algorithm.

4. **Proportions in Admixture** [17]: A linear regression based deconvolution algorithm.

## Spillover to convergence

In cell type deconvolution, spillover refers to the tendency of some cell types to be misclassified as other cell types [18]. For example, when using LM22, deconvolving purified activated mast cell samples results predicted cell compositions that are almost equally split between activated and resting mast cells (Fig 3). One approach to exploring this problem might be to cluster the signature matrix, and assume that highly correlated signatures would tend to spill over to each other. However, **ADAPTS** instead directly calculates what cell types spill over to what other cell types by deconvolving the purified samples, $P$, used to construct and augment the signature matrices, $S$. While the cell types that are likely to spill-over detected by both methods are similar, directly calculating the spillover reveals some surprising patterns. For example, based on signature matrix clustering of LM22, 'Dendritic.cells.activated' and 'Dendritic.cells.resting' tend to cluster together, however the spillover patterns (Fig 4) reveal that 'Dendritic.cells.activated' are most similar to 'Macrophages.M1' while 'Dendritic.cells.resting' are similar to 'Macrophages.M1' and 'Macrophages.M2'. Similarly, T cells are broken into three blocks, implying that if T cells were classified by gene expression rather than surface markers, the broad T cell families (i.e. $CD4^+$ and $CD8^+$) might be differently defined.

As shown in Algorithm 2, recursively (or iteratively) applying the spillover calculation reveals clear clusters of cells. Eq 6 revisits Eq 1, obtaining an initial spillover matrix, $E^0$, by applying Eq 1 to a signature matrix, $S^0$, and the source data used to construct it, $P^0$.

$$E^0 = D(S^0, P^0) \tag{6}$$

Thus $E^0$ would have $|C|$ rows representing each cell type in $S^0$ and one column for each of the $|P^0|$ samples. Applying $A(E^0)$ to average the cell type estimates $E$ across purified samples makes the spillover matrix resemble a signature matrix, leading to Eq 7.

$$S^1 = A(E^0) \tag{7}$$

This new spillover based deconvolution matrix $S^1$ has $|C|$ rows with the average percentage that each of the $|C|$ purified cell types has deconvolved into. $S^1$ can be used to re-deconvolve the initial spillover matrix, $E^0$, effectively 'sharpening' the deconvolution matrix image as shown in Eq 8.

$$E^1 = D(S^1, E^0) \tag{8}$$

Thus $E^1$ will have $|C|$ rows taken from the $|C|$ columns in $S^1$ and $|P^0|$ columns taken from the columns in $E^0$. Once these values are calculated, the following pseudocode (Algorithm 2) shows how **ADAPTS** iteratively applies spillover re-deconvolution to cluster cell types likely to be confused by deconvolution.

**Algorithm 2** Cluster cell types by repeated deconvolution

```
i = 1
while E^i ≠ E^(i-1) do
  i = i + 1
  S^i = A(E^(i-1))
  E^i = D(S^i, E^(i-1))
end while
```

If $D(S, X)$ returns $|C| + 1$ rows, i.e. has an 'others' estimate for cell types not in the matrix, then Eqs 6–8 and Algorithm 2 remain unchanged.

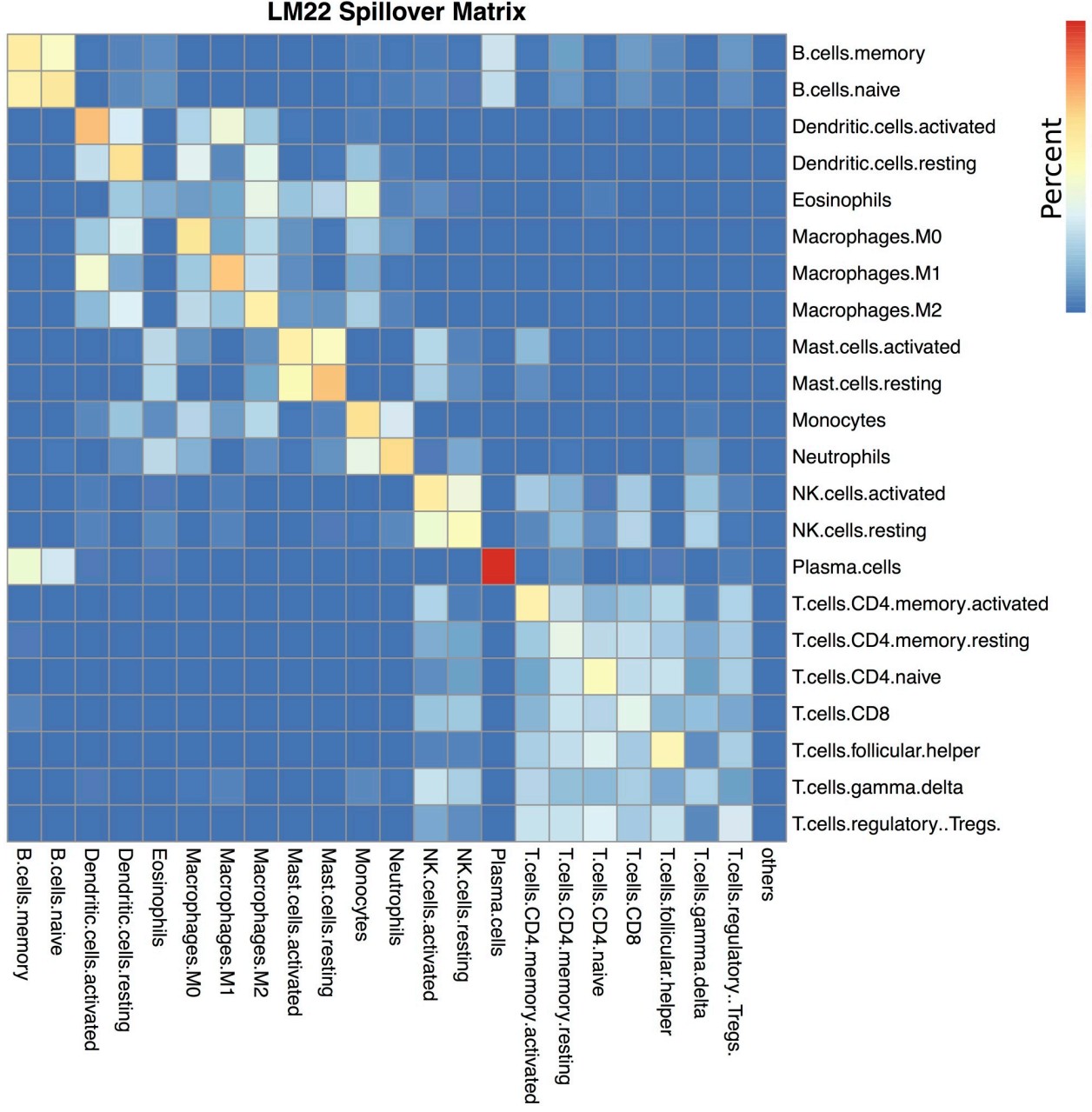

**Fig 3. LM22 spillover matrix.** Spillover matrix showing mean misclassification of purified samples for LM22. Rows show purified cell types and columns show what those samples deconvolve as. Cells are colored by percentage, such that each row adds up to 100%. For example, if the row is 'B.cell. memory', the column is 'Plasma.cells', then the color is light blue indicating that purified 'B.cell.memory' samples deconvolve as containing (on average) 18% 'Plasma.cells'.

As shown in Algorithm 2, the signature matrix may never converge, but instead can alternate between several solutions such that $E^i = E^{i-1}$ is impossible. Therefore **ADAPTS** includes a parameter forcing the algorithm to break and return an answer after $i$ iterations. However, the algorithm usually converges in less than 30 iterations, resulting in a clustered spillover matrix (*e.g.* Fig 4).

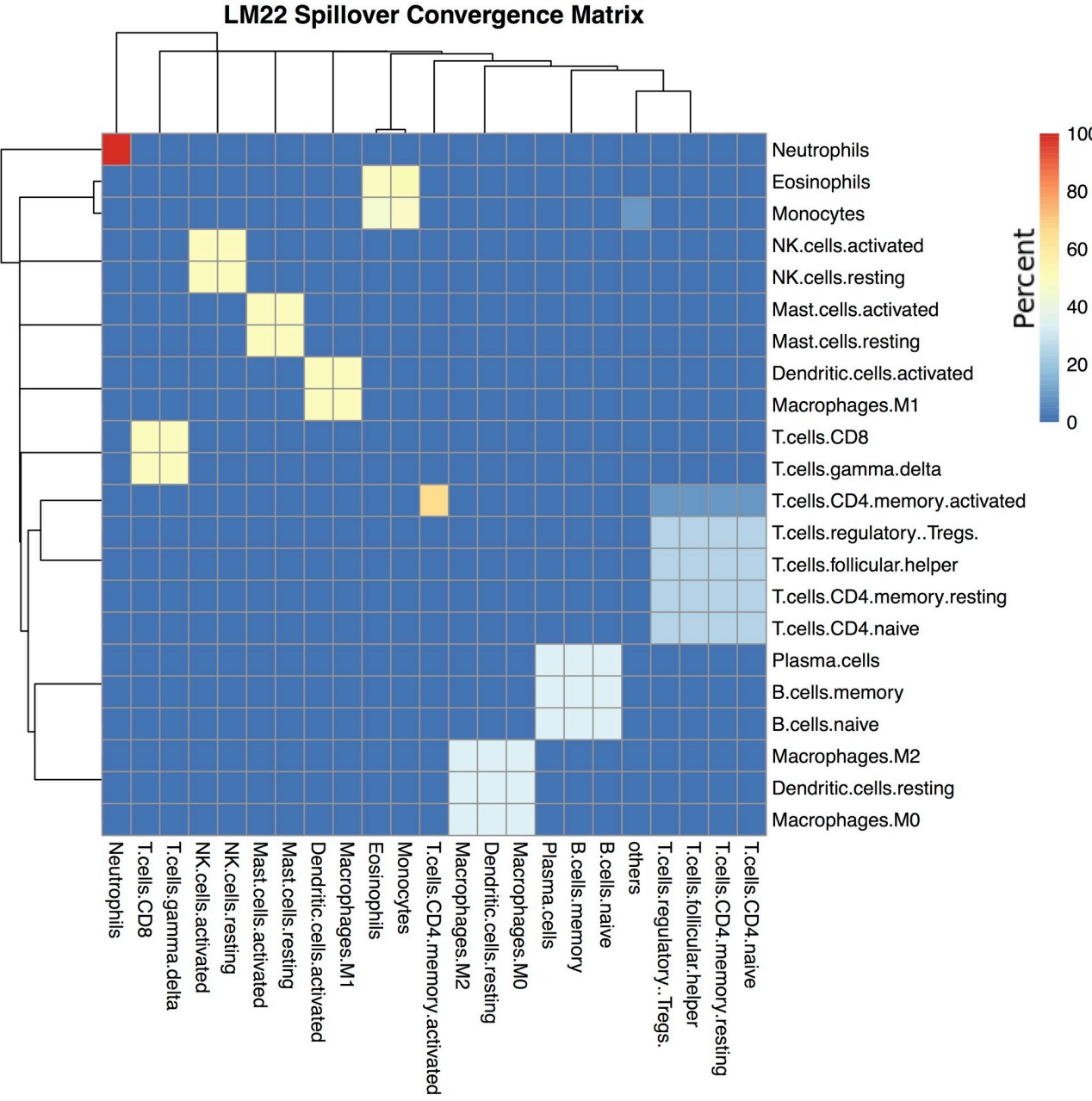

**Fig 4. LM22 converged spillover matrix.** Iterative deconvolution shows how easily confused cell types conspicuously form clusters. Rows show purified cell types and columns show what those samples deconvolve as. Cells are colored by percentage, such that each row adds up to 100%.

The resulting cell type clusters (*CC*) are extracted from $E^i$ by grouping the cell types for any rows that are identical. For example in Fig 4, 'NK.cells.activated' and 'NK.cells.resting' would be grouped in one cluster (e.g. $CC^3$), while 'Neutrophils' would exist in a cluster by themselves ($CC^2$), and $|CC| = 10$.

Cell types co-clustered by this method are cell types that have similar deconvolution patterns from purified samples. Within cell type clusters a cell type will usually deconvolve most frequently as itself and less frequently as other cell types within the cluster (e.g. NK.cells.activated and NK.cells.resting in Figs 3 and 4). Thus clusters are not necessarily a block within

which cellular proportions are inaccurate. Rather, co-clustered cell types would benefit from additional efforts to separate them, such as by finding genes that specifically differentiate those cell types as is done in hierarchical deconvolution.

## Hierarchical deconvolution

The clusters calculated by Algorithm 2 allow the hierarchical deconvolution implemented in **ADAPTS**. **ADAPTS** includes a function to automatically train deconvolution matrices that include only genes that differentiate cell types that cluster together in Algorithm 2. The first round of deconvolution determines the total fraction of cells in the cluster. The next round of deconvolution determines the relative proportion of all of the cell types in that cluster as shown in Algorithm 3.

While this algorithm has not been implemented recursively in **ADAPTS**, if it was it would resemble a discrete version of the continuous model implemented in MuSiC [19].

**Algorithm 3** Hierarchical deconvolution

```
Require: CC, Pⁿᵉʷ, S⁰, Sᴱ⁰, and P {Pⁿᵉʷ has |G| genes × |J| new samples
to predict}
    Sᵇᵃˢᵉ = Algorithm 1(S⁰, Sᴱ⁰, P) {|g| genes × |C| cell types}
    Eᵇᵃˢᵉ = D(Sᵇᵃˢᵉ, Pⁿᵉʷ) {|C| cell types × |J| samples from Pⁿᵉʷ}
    for cc ∈ CC do
        EB = ∑_{c'∈cc} E^{base}_{c',}
        varsᶜᶜ = ∀g ∈ G : variance(P^{cc}_g)
        seedSize = ⌈nrow(Sᵇᵃˢᵉ)/10⌉
        gᶜᶜ = seedSize genes with the top values in varsᶜᶜ
        Sᶜᶜ = Algorithm 1(A(P^{cc}_{g^{cc}}), Pᶜᶜ, Pᶜᶜ)
        EC = D(Sᶜᶜ, Pⁿᵉʷ)
        for c ∈ cc do
            Eᶜ = EB × (EC_{c,} / ∑_{c'∈cc} EC_{c',}) {1 cell type × |J| samples in Pⁿᵉʷ}
        end for
    end for
    Eⁿᵉʷ = ((E^{c₁})ᵀ|...|(E^{c_{|C|}})ᵀ)ᵀ {|C| cell types × |J| samples from Pⁿᵉʷ}
```

## Evaluation metrics

To evaluate the efficacy of deconvolution, we use two common metrics: Pearson's correlation coefficient ($\rho$) and root mean squared error (RMSE). Correlation represents the quality of the linear relationship between the predicted and actual cell type percentages. Root mean square error (RMSE) directly measures the error between the actual and predicted values for a cell type. A detailed discussion of these metrics and other alternatives are presented by Li and Wu [20].

In Example 1: Detecting Tumor Cells, correlation and RMSE evaluate the accuracy of predicting a single cell type (i.e. myeloma tumor) across multiple samples. In this kind of analysis, if a particular cell type has a high correlation, but also a high RMSE, that indicates that the estimates are systematically underestimating (or overestimating) the actual percentage of that cell type. In the case of underestimation, some percentage of that cell type is being misclassified as another cell type or cell types (and *vice versa* for overestimation).

In Example 2: Deconvolving Single Cell Pancreas Samples, correlation and RMSE evaluate predictions for all cell types in a single sample. In this case, the aforementioned bias is not possible since both the predicted and actual cell percentages must add up to 100%.

## Results

The following results section shows how the theory set out in Materials and Methods is applied to detect tumor cells in multiple myeloma samples and to utilize single cell RNAseq data to build a new signature matrix. It contains highlights from two vignettes distributed with the CRAN package (S1 and S2 Vigs).

### Example 1: Detecting tumor cells

To demonstrate utility of the **ADAPTS** package, we show how it can be used to augment the LM22 from [5] to identify myelomatous plasma cells from gene expression profiles of 423 purified tumor (CD138[+]) samples and 440 whole bone marrow (WBM) samples taken from multiple myeloma patients. The fraction of myeloma cells, which are tumorous plasma cells, were identified in both sample types via quantification of the cell surface marker CD138. Root mean squared error (RMSE) and Pearson's correlation coefficient ($\rho$) were used to evaluate accuracy of tumor cell fraction estimates. RMSE proved particularly relevant when deconvolving purified CD138[+] sample profiles, because 356 of 423 samples are more than 90% pure tumor resulting in clumping of samples with purity near 100%.

The following matrices were used or generated during the evaluation:

- **LM22**: As reported in [5]. The sum of the 'memory B cells' and 'plasma cells' deconvolved estimates represent tumor percentage.

- **LM22 + 5**: Builds on LM22 by adding purified sample profiles for myeloma specific cell types as shown in Eq 2: plasma memory cells [21], osteoblasts [9], osteoclasts, adipocytes, and myeloma plasma cells [22]. The sum estimates for 'memory B cells', 'myeloma plasma cells', 'plasma cells', and 'plasma memory cells' represent tumor percentage.

- **MGSM27**: Builds on LM22 by adding 5 myeloma specific cell types using **ADAPTS** to determine inclusion of additional genes as shown in Eq 3. Fig 2 shows **ADAPTS** evaluating matrix stability after adding different numbers of genes, smoothing the condition numbers, and selecting an optimal number of features.

- ***de novo* MGSM27**: Builds a *de novo* MGSM27 by seeding with the 100 most variable genes from publicly available data similar to those mentioned in [5] and the 5 aforementioned myeloma specific cell types.

Table 1 displays average *RMSE* and $\rho$ for tumor fraction estimates obtained via application of DCQ deconvolution using the four aforementioned matrices across both myeloma profiling datasets.

While the exact genes chosen during each run varies slightly, Table 1 shows that consistently the best accuracy is achieved by augmenting LM22 using **ADAPTS**. The reduced

**Table 1. Deconvolved tumor accuracy in WBM samples.**

| Matrix | WBM *RMSE* | WBM $\rho$ | CD138[+] *RMSE* | CD138[+] $\rho$ |
|---|---|---|---|---|
| LM22 | 38.0 ± 0.0 | 0.59 ± 0.00 | 27.0 ± 0.0 | 0.26 ± 0.00 |
| LM22 + 5 | 37.0 ± 0.0 | 0.53 ± 0.00 | 12.0 ± 0.0 | 0.26 ± 0.00 |
| MGSM27 | **23.0 ± 0.0** | 0.60 ± 0.00 | **09.0 ± 0.2** | **0.33 ± 0.01** |
| *de novo* MGSM27 | 24.0 ± 0.0 | **0.65 ± 0.00** | 36.0 ± 0.0 | 0.18 ± 0.00 |

Deconvolution reconstruction of tumor percentage in whole bone marrow (*WBM*) and samples sorted to consist of nearly pure *CD138*[+] cells. Classifier accuracy is measured by root mean square error (*RMSE*) and Pearson's correlation coefficient ($\rho$). The best scores in each column are bolded.

**Table 2. Deconvolution of CD138+ purified samples.**

| Cell Type | LM22 | LM22p5 | MGSM27 | *de novo* MGSM27 |
|---|---|---|---|---|
| B.cells.naive | 1.76 | 0.92 | 0.2 | 0 |
| B.cells.memory* | 10.38 | 5.27 | 2 | 0.02 |
| Plasma.cells* | 59.59 | 45.49 | 31.77 | 7.58 |
| T.cells.CD8 | 1.06 | 0.13 | 0 | 0 |
| T.cells.CD4.naive | 2.58 | 0.09 | 0.49 | 4.94 |
| T.cells.CD4.memory.resting | 1.33 | 0.13 | 0 | 0 |
| T.cells.CD4.memory.activated | 0.03 | 0.01 | 0 | 0.01 |
| T.cells.follicular.helper | 5.32 | 0.36 | 0.9 | 4.74 |
| T.cells.regulatory..Tregs. | 2.72 | 0.35 | 1.14 | 7.37 |
| T.cells.gamma.delta | 0.03 | 0.03 | 0.06 | 6.89 |
| NK.cells.resting | 0.11 | 0.09 | 0.03 | 0 |
| NK.cells.activated | 0.03 | 0.01 | 0 | 0 |
| Monocytes | 5.57 | 1.13 | 0.57 | 0.05 |
| Macrophages.M0 | 1.36 | 0.11 | 0.03 | 0 |
| Macrophages.M1 | 1.06 | 0.08 | 0.04 | 0.02 |
| Macrophages.M2 | 4.14 | 0.49 | 0.52 | 0.13 |
| Dendritic.cells.resting | 0.95 | 0.11 | 0.05 | 0 |
| Dendritic.cells.activated | 0.57 | 0.08 | 0.06 | 0 |
| Mast.cells.resting | 0.3 | 0.14 | 0.01 | 0.09 |
| Mast.cells.activated | 0.06 | 0.03 | 0 | 0.01 |
| Eosinophils | 0.72 | 0.21 | 0.46 | 7.24 |
| Neutrophils | 0.32 | 0.29 | 0.11 | 0.01 |
| others | 0 | 0 | 0 | 0 |
| PlasmaMemory* | 0 | 32.56 | 24.07 | 21.76 |
| MM.plasma.cell* | 0 | 4.62 | 34.51 | 29.12 |
| osteoclast | 0 | 0.8 | 2.93 | 4.54 |
| osteoblast | 0 | 1.1 | 0.02 | 2.93 |
| adipocyte | 0 | 5.37 | 0.01 | 2.55 |

Average (mean) deconvolution reconstruction of all cell types in the 423 nearly pure CD138+ samples.

*Cell types are counted as CD138+ tumor

performance of the *de novo* MGSM27 on the CD138+ samples is likely due to genes that were present in LM22, but were missing in some of the source data and thus excluded from *de novo* construction. More details are available in the vignette distributed with the **R** package.

**Deconvolved cell types.** Cell type estimates of non-CD138+ cells from immunostaining are not available for the 423 CD138+ or 440 WBM samples. However, enumerating the cell types detected by deconvolution in the CD138+ and WBM samples by the four signature matrices illustrates the changes brought about by augmenting the signature matrix. Table 2 shows the mean percentage of each cell type across the 423 CD138+ samples and Table 3 shows the same for the 440 WBM samples.

In the CD138+ samples (Table 2), adding 5 cell types without adding any additional genes has the immediate benefit of greatly reducing the percentage of cells classed as immune cells that should not be in the sample (e.g. 'T.cells.follicular.helper', 'Monocytes'). However, this is balanced out by adding percentages of new cell types that should not be in the samples (e.g. 'adipocyte') and deconvolving only a relatively small percentage (4.62%) as the most obvious tumor type: 'MM.plasma.cell'. This is consistent with the hypothesis that the signature matrix

**Table 3. Deconvolution of WBM samples.**

| Cell Type | LM22 | LM22p5 | MGSM27 | *de novo* MGSM27 |
|---|---|---|---|---|
| B.cells.naive | 0.12 | 0.15 | 0.01 | 0 |
| B.cells.memory* | 0.84 | 0.55 | 0.09 | 0.01 |
| Plasma.cells* | 9.84 | 4.59 | 3.89 | 0.85 |
| T.cells.CD8 | 1.81 | 1.15 | 0.05 | 0 |
| T.cells.CD4.naive | 0.59 | 0.14 | 0.12 | 3.99 |
| T.cells.CD4.memory.resting | 0.24 | 0.15 | 0 | 0 |
| T.cells.CD4.memory.activated | 0 | 0 | 0 | 0 |
| T.cells.follicular.helper | 0.18 | 0.02 | 0.02 | 3.65 |
| T.cells.regulatory..Tregs. | 0.16 | 0.06 | 0.06 | 3.33 |
| T.cells.gamma.delta | 3.53 | 4.22 | 3.25 | 7.14 |
| NK.cells.resting | 11 | 11.73 | 4.62 | 0.27 |
| NK.cells.activated | 1.48 | 1.54 | 0.1 | 0 |
| Monocytes | 14.37 | 12.94 | 8.12 | 4.09 |
| Macrophages.M0 | 4.75 | 2.24 | 1.83 | 0.86 |
| Macrophages.M1 | 2.09 | 1.07 | 1.11 | 1.15 |
| Macrophages.M2 | 7.63 | 4.81 | 5 | 4.18 |
| Dendritic.cells.resting | 0.73 | 0.23 | 0.17 | 0.02 |
| Dendritic.cells.activated | 0.27 | 0.19 | 0.07 | 0.01 |
| Mast.cells.resting | 12.11 | 12.4 | 4.9 | 0 |
| Mast.cells.activated | 6.76 | 6.73 | 1.57 | 0 |
| Eosinophils | 7.23 | 7.44 | 7.08 | 13.23 |
| Neutrophils | 14.29 | 15.63 | 8.04 | 4.05 |
| others | 0 | 0 | 0 | 0 |
| PlasmaMemory* | 0 | 6.19 | 11.2 | 11.51 |
| MM.plasma.cell* | 0 | 1.38 | 14.78 | 16.82 |
| osteoclast | 0 | 2.27 | 14.37 | 13.95 |
| osteoblast | 0 | 0.32 | 5.92 | 4.54 |
| adipocyte | 0 | 1.86 | 3.62 | 6.34 |

Average (mean) deconvolution percentages of all cell types in the 440 WBM samples.

*Cell types are counted as $CD138^+$ tumor

lacks the correct genes to precisely identify these cell types. Adding in additional genes to make MGSM27 moves cancer types 'MM.plasma.cell', 'Plasma.cells', and 'PlasmaMemory' cells to the most frequent cell types and reduces all other cell types to very low estimates. The *de novo* MGSM27 shows an increase in many cell types that should not be in the samples. One explanation for this might be that the genes common to 'B.cells.memory', 'Plasma.cells', 'PlasmaMemory', and 'MM.plasma.cell' are scored inappropriately lowly by Eq 4.

In the WBM samples (Table 3), adding 5 cell types without adding any additional genes results in little change to the average percentage of cell types excepting the reduction in 'Plasma.cells' percentage which apparently shift to 'PlasmaMemory'. Adding in additional genes to make MGSM27 greatly increases the estimates for the new stromal cell types (i.e. 'osteoclasts', 'osteoblasts', and adipocytes) as well as the tumor cell types largely at the expense of innate immune cells (e.g. Monocytes, Mast.cells, Neutrophils, *et al*.). This removes a bias where tumor cells were systematically under-represented (12.71% in LM22p5 versus 29.96% in MGSM27). The *de novo* MGSM27 results are similar to MGSM27, except for an increase in certain types of T cells and a shifting in innate populations.

**Spillover matrix.**   Successfully recapturing the known percentage of tumor cells in a sample is a useful intermediate validation step, however, the true value of a deconvolution algorithm lies in its ability to determine cell types in a sample that affect patient outcomes. Statistical and machine learning techniques may be applied to identify relevant cell estimates. From there, a correct understanding of the limitations of deconvolution is helpful to reveal the underlying biology. One particularly relevant limitation of deconvolution is how the algorithm may confuse different cell types. **ADAPTS**'s approach to resolving this problem is detailed in the Materials and Methods subsection Spillover to Convergence producing spillover matrices such as Fig 3 that show what samples of a single purified cell type deconvolve as.

Recursive application of the spillover concept leads to Algorithm 2 and cell type clusters such as those shown in Fig 4. One way to interpret these results is that co-clustered cell types are those cannot be reliably distinguished by deconvolution using a particular deconvolution algorithm and signature matrix. In this example, 'B.cells.naive', 'B.cells.memory', and 'Plasma. cells' are all clearly clustered together. These clusters may be particularly valuable for single cell RNAseq analysis where clustering software such as Seurat [23] aid in annotating cell types, but can introduce artificial distinctions due to limitations inherent in clustering.

## Example 2: Deconvolving single cell pancreas samples

In this section we demonstrate how **ADAPTS** can be applied to build a deconvolution matrix from single cell RNAseq data. This example has the additional benefit of illustrating the utility of the algorithms outlined in Spillover to Convergence and Hierarchical Deconvolution to find cell type clusters and distinguish between cell types in those clusters. In this example we use the pancreas single cell RNAseq dataset available in Array Express [9] as E-MTAB-5061 [24]. All cells of single type were combined and averaged to build pseudo-pure samples of each annotated cell types. A pseudo-bulk RNAseq sample was constructed by adding together all cell types, with the pseudo-bulk cell type percentages assigned based on the proportion of annotated single cells in the mix. The normal pancreas samples were used as the training set and the diabetic pancreas samples as the test set.

To demonstrate the utility of augmenting a signature matrix with **ADAPTS**, we build a signature matrix from the top 100 most variant genes (i.e. Top100) and then augmented this signature (i.e Augmented) as shown in Fig 5. The first test is to predict the normal pseudo-bulk data—essentially predicting the training set (Table 4). The second test is a blind estimation of the diabetic pancreas sample (Table 5). As shown in Table 4 the Top100 genes set the baseline correlation coefficient (i.e. $\rho$) at 0.05 and the root mean square error (*RMSE*) at 13.82. Augmenting the signature matrix with **ADAPTS** Algorithm 1 improved the rho to 0.26 and RMSE to 10.72.

**Clustering cell types improves deconvolution accuracy.**   The spillover clustering algorithm outlined in section was applied to the Top100 and Augmented signature matrices. Fig 6 shows the cell type clusters for the Top100 signature matrix, and Fig 7 for the Augmented signature matrix. One way to interpret the results is to assume that the clustered cell types are indistinguishable from each other, then the correct comparison method is to treat both as the same cell type. Combining the clustered cell types for the Top100 estimates increased the $\rho$ to 0.32 but also increased the *RMSE* to 17.15. Similarly, the Augmented cell estimates had $\rho$ = 0.58 and *RMSE* = 16.58. Using Spearman's rank-order correlation, we saw only modest increases in the correlation coefficient: the Top100 Spearman's $\rho$ increased from 0.50 to 0.51 and the Augmented Spearman's $\rho$ increased 0.69 to 0.71. Thus the bulk of the improvement in

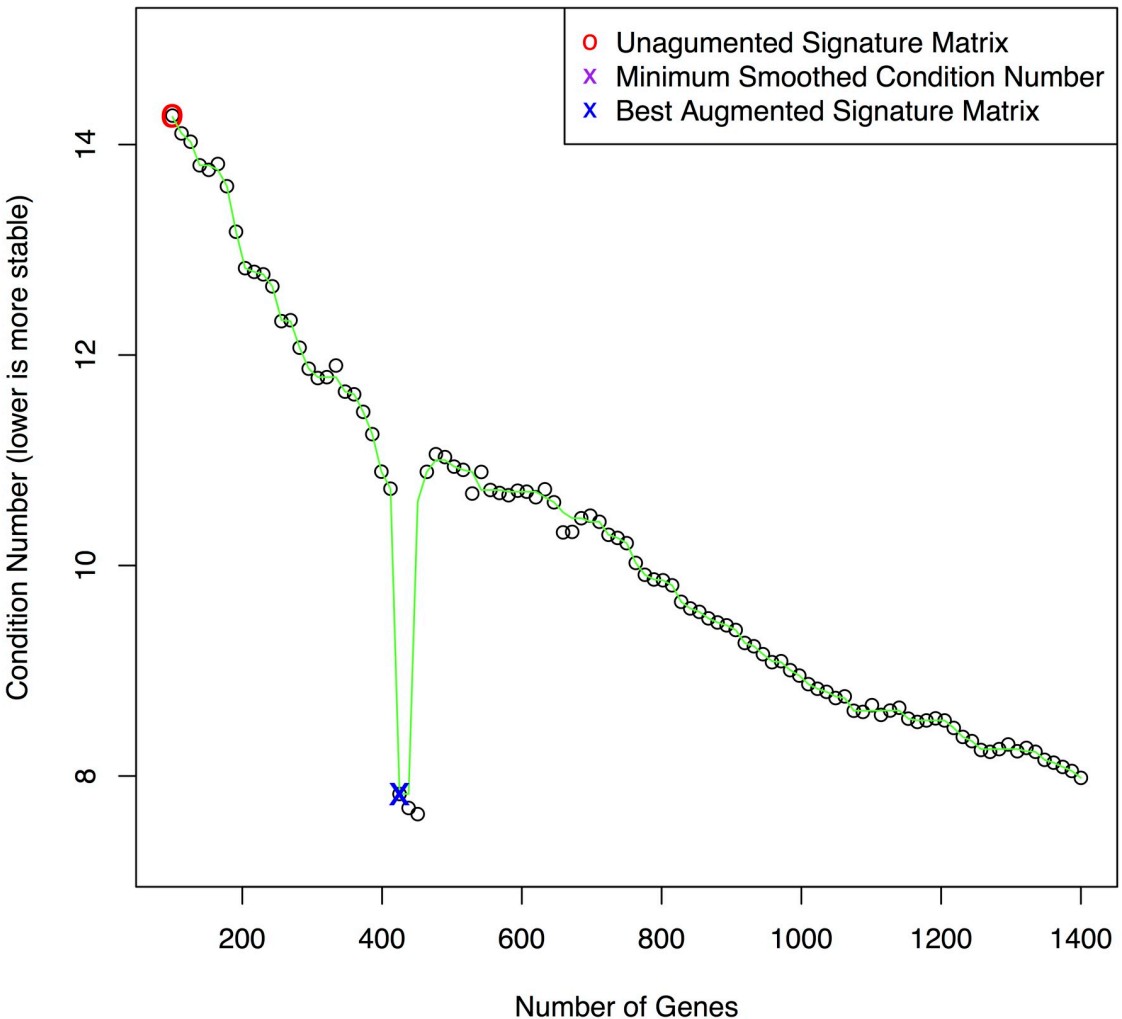

**Fig 5. scRNAseq signature matrix construction.** Curve showing the selection of an optimal condition number for the single cell RNAseq augmented signature matrix data.

Pearson's correlation came from effectively removing from consideration low abundance cell types that were estimated as 0% using the Top100 matrix or being over-estimated from the spillover of other cell types using the Augmented matrix.

**Hierarchical clustering improves deconvolution accuracy.** ADAPTS Algorithm 3 (outlined in Hierarchical Deconvolution) was used to build custom signature matrices for breaking apart the clusters shown in Figs 6 and 7. This improved deconvolution accuracies shown in the 'hierarchical' columns of Table 4. Applying the model built on the normal samples to the diabetic pancreas resulted in the even better blind predictive accuracies shown in Table 5 with the overall best accuracy provided by the hierarchical deconvolution using the Augmented signature matrix: $\rho = 0.46$, $RMSE = 8.91$.

**Table 4. Deconvolution of pancreas training set.**

| Cell Type | Top 100 | Top 100 hierarchical | Augmented | Augmented hierarchical | Reference |
|---|---|---|---|---|---|
| acinar.cell | 11.38 | 7.88 | 11.56 | 10.49 | 10.36 |
| ductal.cell | 7.32 | 7.85 | 7.85 | 12.56 | 43.11 |
| alpha.cell | 7.46 | 7.92 | 8.34 | 9.51 | 12.11 |
| gamma.cell | 11.66 | 7.77 | 9.68 | 8.51 | 1.83 |
| beta.cell | 7.11 | 11.75 | 7.66 | 10.77 | 3.51 |
| co.expression.cell | 4.36 | 16.91 | 11.49 | 8.41 | 14.26 |
| delta.cell | 0.00 | 12.27 | 2.56 | 8.04 | 0.88 |
| unclassified.endocrine.cell | 7.02 | 20.63 | 6.11 | 12.29 | 0.40 |
| endothelial.cell | 8.37 | 0.00 | 7.63 | 0.00 | 8.53 |
| PSC.cell | 0.00 | 0.00 | 2.13 | 8.52 | 0.32 |
| epsilon.cell | 0.00 | 7.02 | 2.68 | 6.11 | 0.16 |
| mast.cell | 0.00 | 0.00 | 5.96 | 0.80 | 2.55 |
| MHC.class.II.cell | 35.33 | 0.00 | 16.37 | 4.01 | 1.99 |
| others | 0.00 | 0.00 | 0.00 | 0.00 | 0.00 |
| *RMSE* | 13.82 | 12.09 | 10.72 | 10.16 | 0.00 |
| $\rho$ | 0.05 | 0.12 | 0.26 | 0.39 | 1.00 |

Deconvolution cell type estimates of the normal pancreas training set.

## Discussion

In previous sections, we demonstrate that **ADAPTS** is a useful tool for automating the addition of additional cell types to a signature matrix and for performing hierarchical deconvolution to improve the accuracy of cell type deconvolution. Here we discuss extensions the **ADAPTS** workflow (as presented in the Vignettes) to other deconvolution related problems as

**Table 5. Deconvolution of pancreas test set.**

| Cell Type | Top 100 | Top 100 hierarchical | Augmented | Augmented hierarchical | Reference |
|---|---|---|---|---|---|
| acinar.cell | 7.40 | 4.32 | 10.82 | 8.89 | 5.78 |
| alpha.cell | 7.93 | 9.01 | 7.94 | 14.41 | 36.24 |
| beta.cell | 8.04 | 8.44 | 8.58 | 9.35 | 12.39 |
| co.expression.cell | 12.33 | 7.95 | 10.03 | 8.55 | 1.68 |
| delta.cell | 9.38 | 11.50 | 8.13 | 10.93 | 7.35 |
| ductal.cell | 5.93 | 17.06 | 12.49 | 8.90 | 21.74 |
| endothelial.cell | 0.00 | 13.80 | 2.58 | 7.91 | 0.53 |
| epsilon.cell | 6.56 | 21.36 | 5.83 | 12.12 | 0.21 |
| gamma.cell | 8.46 | 0.00 | 7.61 | 0.00 | 9.45 |
| mast.cell | 0.00 | 0.00 | 1.72 | 8.51 | 0.32 |
| MHC.class.II.cell | 0.00 | 6.56 | 2.88 | 5.83 | 0.32 |
| PSC.cell | 0.00 | 0.00 | 5.93 | 0.55 | 2.31 |
| unclassified.endocrine.cell | 33.97 | 0.00 | 15.47 | 4.05 | 1.68 |
| others | 0.00 | 0.00 | 0.00 | 0.00 | 0.00 |
| *RMSE* | 12.76 | 10.68 | 9.44 | 8.91 | 0.00 |
| $\rho$ | 0.06 | 0.24 | 0.35 | 0.46 | 1.00 |

Blind deconvolution cell type estimates of the diabetic pancreas test set.

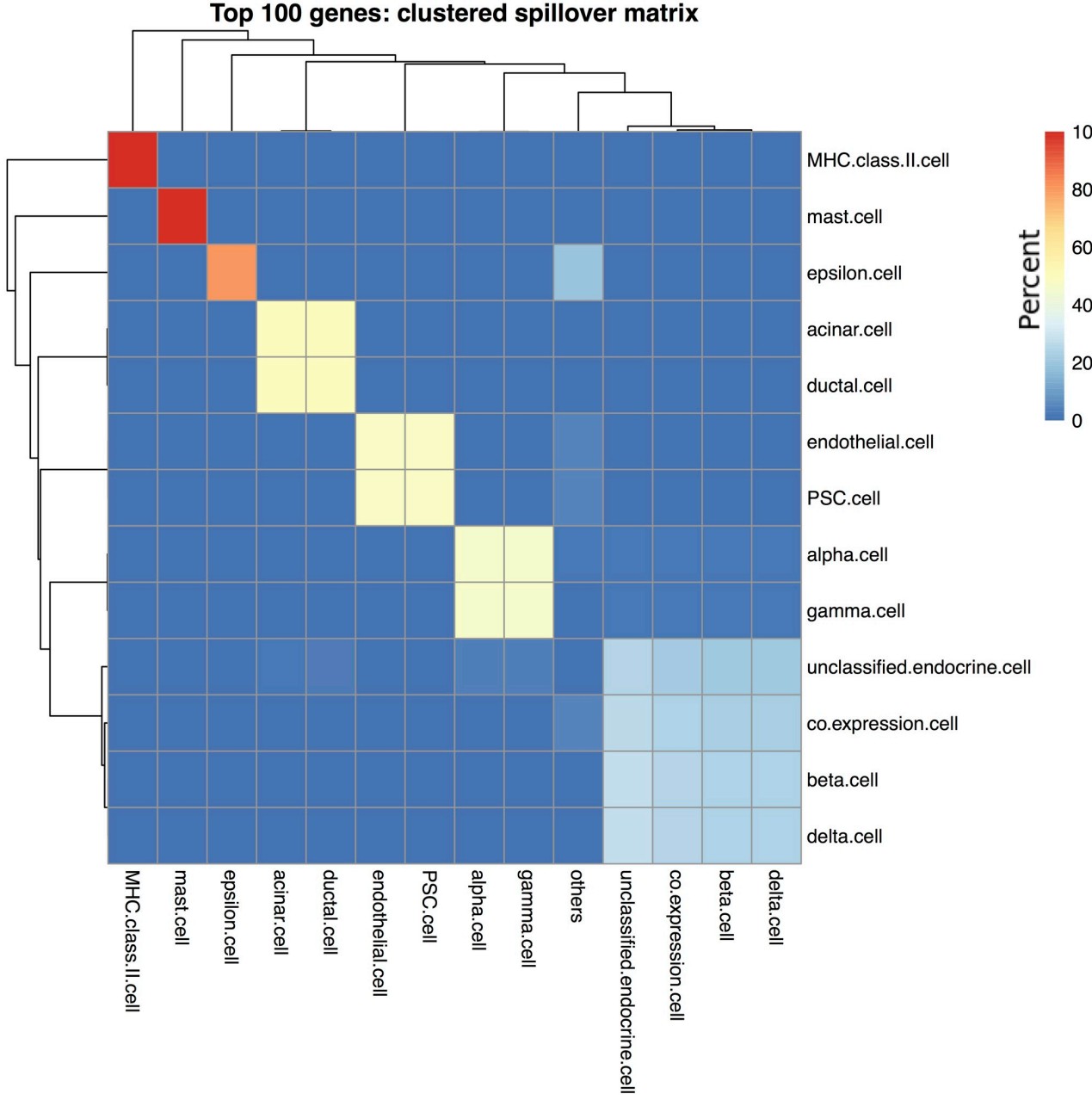

**Fig 6. Clustering of Top 100 gene signature matrix.** The cell type clusters identified using the signature matrix constructed from the 100 genes with the highest variance across cell types in the single cell data drawn from a normal pancreas sample. Rows show purified cell types and columns show what those samples deconvolve as. Cells are colored by percentage, such that each row adds up to 100%.

well as potential future modifications to the core package may improved ADAPTS-based deconvolution.

## Cancer suppressed immune and stromal cells

Most gene expression data generated from purified immune cells, and thus most publicly available signature matrices, are generated from peripheral blood. However, immune and stromal cells in a tumor microenvironment are expected to differ in gene expression from those

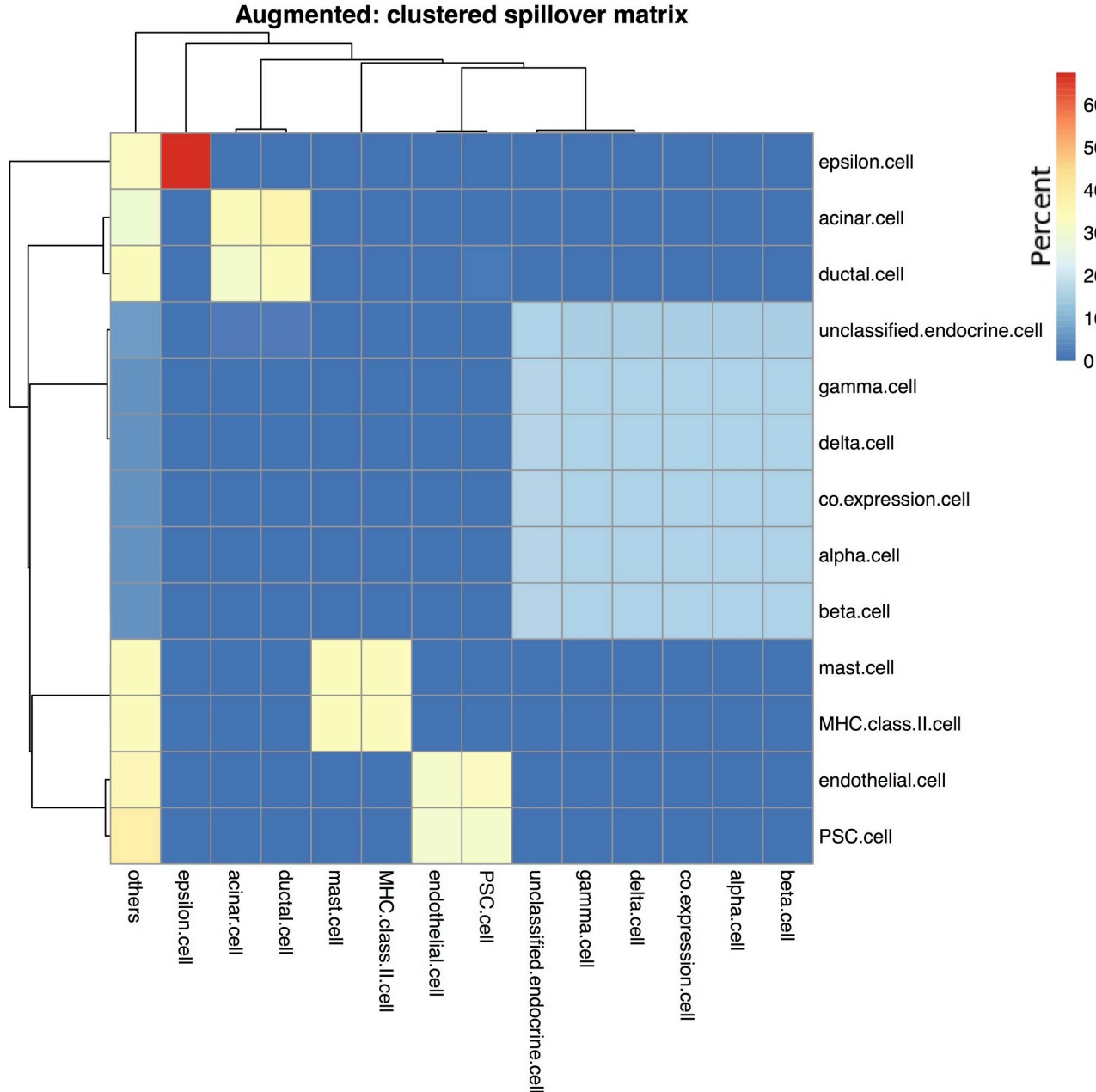

**Fig 7. Clustering of augmented gene signature matrix.** The cell type clusters identified using the augmented signature matrix that was seeded with the 100 genes exhibiting the highest variance in the normal pancreas sample. Rows show purified cell types and columns show what those samples deconvolve as. Cells are colored by percentage, such that each row adds up to 100%.

purified from non-tumor sources [25]. If gene expression of cells purified from the relevant tumor microenvironment is available, it would be straight-forward to use that data in **ADAPTS**. Either the tumor-associated cell types could be used to augment an existing signature matrix, or the relevant cell type (and the top genes for that cell type) could be removed from the signature matrix before **ADAPTS** augments the cell type back in to the signature matrix using the new data.

If a tumor microenvironment sample has been subjected to a scRNAseq experiment, then this offers the potential to build a more accurate signature matrix than one from any purified cell type data. Individual cells will be assigned to clusters which roughly correlate with purified cell types. Theoretically, these cell type clusters would represent the immune, stromal, and tumor cell gene expression in the microenvironment and enumerate exactly the cell types present in the sample. See Example 2: Deconvolving Single Cell Pancreas Samples for an example of how to construct a signature matrix *de novo* from single cell data using **ADAPTS**.

### Co-linear cell types

In Example 1: Detecting Tumor Cells, tumor cells are represented by four cell types with very similar gene expression patterns and this apparently creates problems, especially in *de novo* signature matrix construction. Eq 4 scores rank genes for each cell type to be iteratively added to the signature matrix by Algorithm 1. When Eq 4 scores a set of cell types where a subset of the cell types have similar gene expression, it may underscore genes that are helpful for differentiating that subset from all other cell types.

One might imagine algorithms that would potentially improve this situation by either increasing the score of genes common to subsets of cell types, or make certain that these genes appear in the seed matrix. For instance, cell types that are highly co-linear (as detected by the spill-over clustering or other method) might be pooled together into a common cell type for purposes of determining gene scores with the **ADAPTS** function **rankByT**. While high variance genes may already be present in the seed matrix, perhaps the gene scores should be up-weighted for genes that have a globally high variance; this would improve the representation of genes that are high in all tumor cell types (for example), but low in all others. Or perhaps genes might be scored in **rankByT** by running pair-wise comparisons between all cell types, and the genes score for a particular cell type might be its top ratio against *any* other cell type rather than *all* other cell types.

Ultimately, solving this problem is beyond the scope of this publication, however the **ADAPTS** framework can help to test potential solutions which may then be shared via **GitHub** and potentially included in the official **CRAN** package.

### Conclusion

Table 1 shows an example where using **ADAPTS** to include additional genes and tissue specific cell types improves the ability of a deconvolution algorithm to identify tumor fractions in microarray-based purified and mixed multiple myeloma gene expression samples. Thus we demonstrate that the techniques implemented in **ADAPTS** are potentially beneficial for many situations. The functions implemented in **ADAPTS** enable researchers to build their own custom signature matrices and investigate biosamples consisting of multiple cell types. Tables 4 and 5 show that these methods can build new signature matrices from single cell RNAseq (scRNAseq) data and effectively deconvolve the cell types determined by single cell analysis. This is expected to be particularly useful as researchers use scRNAseq to determine cell types that are present in tissue where large numbers of bulk gene expression samples are already available.

### Supporting information

**S1 Vig. ADAPTS (Automated deconvolution augmentation of profiles for tissue specific cells) Vignette.**
(HTML)

**S2 Vig. ADAPTS Vignette 2: Single cell analysis.**
(HTML)

## Acknowledgments

Thanks to Gareth Morgan, Jake Gockley, Robert Hershberg, Mary H Young, Andrew Dervan, and all other contributors to the paper "Characterizing bone marrow microenvironments which contribute to patient outcomes in newly diagnosed multiple myeloma". We would also like to thank Xiling Shen and Wennan Chang for taking the time to provide thoughtful critiques that improved the readability of this manuscript and contributed substantially to the discussion.

## Author Contributions

**Conceptualization:** Samuel A. Danziger, Mark McConnell, Frank Schmitz, Alexander V. Ratushny.

**Data curation:** Samuel A. Danziger.

**Formal analysis:** Samuel A. Danziger, Alexander V. Ratushny.

**Investigation:** Samuel A. Danziger.

**Methodology:** Samuel A. Danziger, Frank Schmitz, David J. Reiss, Alexander V. Ratushny.

**Project administration:** Samuel A. Danziger.

**Resources:** Samuel A. Danziger.

**Software:** Samuel A. Danziger, David L. Gibbs, Mark McConnell.

**Supervision:** Ilya Shmulevich, Matthew W. B. Trotter, Alexander V. Ratushny.

**Validation:** Samuel A. Danziger.

**Visualization:** Samuel A. Danziger.

**Writing – original draft:** Samuel A. Danziger, Matthew W. B. Trotter.

**Writing – review & editing:** Samuel A. Danziger, Alexander V. Ratushny.

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
