## [Decision Letter · Decision Letter 0]

30 Aug 2019

PONE-D-19-19603

ADAPTS: Automated Deconvolution Augmentation of Profiles for Tissue Specific cells

PLOS ONE

Dear Dr. Danziger,

Thank you for submitting your manuscript to PLOS ONE. After careful consideration, we feel that it has merit but does not fully meet PLOS ONE’s publication criteria as it currently stands. Therefore, we invite you to submit a revised version of the manuscript that addresses the points raised during the review process.

ADAPTS presented a user friendly modular tool that aims to solve a most challenging bioinformatics problem that has huge potential in many fields. The augmentation of signature matrix by custom data is a great idea, however, as the reviewers suggest, issues also come along. 

We would appreciate receiving your revised manuscript by Oct 14 2019 11:59PM. To enhance the reproducibility of your results, we recommend that if applicable you deposit your laboratory protocols in protocols.io, where a protocol can be assigned its own identifier (DOI) such that it can be cited independently in the future. For instructions see: http://journals.plos.org/plosone/s/submission-guidelines#loc-laboratory-protocols

We look forward to receiving your revised manuscript.

Kind regards,

Sha Cao, Ph.D

Academic Editor

PLOS ONE

Journal Requirements:

2. Please amend either the abstract on the online submission form (via Edit Submission) or the abstract in the manuscript so that they are identical.

The authors(s) received no specific funding for this work.

We note that one or more of the authors are employed by a commercial company: Institute for Systems Biology and Celgene.

Reviewer's Responses to Questions

**Comments to the Author**

1. Is the manuscript technically sound, and do the data support the conclusions?

Reviewer #1: Yes

Reviewer #2: Yes

2. Has the statistical analysis been performed appropriately and rigorously? 

Reviewer #1: Yes

Reviewer #2: Yes

3. Have the authors made all data underlying the findings in their manuscript fully available?

Reviewer #1: Yes

Reviewer #2: Yes

4. Is the manuscript presented in an intelligible fashion and written in standard English?

Reviewer #1: Yes

Reviewer #2: Yes

5. Review Comments to the Author

Reviewer #1: Cell type deconvolution is an innovative technique to achieve higher resolution from bulk transcriptomic data. In this manuscript, Danziger, et al., build on existing methods for profiling leukocyte infiltration of tumors by offering their package, ADAPTS, as a method to augment the reference signature matrix used to deconvolve bulk, heterogenous samples. The authors aimed to address two issues that plague implementation of deconvolution algorithms: the presence of stromal cells alongside immune cells that complicates analysis and the tendency for misclassification of closely related cell types, which they termed spillover. The ADAPTS package enables augmentation of existing signature matrices by including differentially expressed genes which define gene expression profiles for additional cell types and minimizes condition number, a measure of matrix stability. The ADAPTS package deals with spillover by iteratively performing deconvolution on highly correlated cell types then applies hierarchical deconvolution on a subset of the genes that define each cell type. The ADAPTS package aims to offer personalization of existing deconvolution techniques to an individual study, but the added value demonstrated in the included examples is questionable. Some clarification of the existing evidence is required.

Major:

1. Using the top 100 differentially expressed genes on average between the existing signature matrix cell types and the added pancreatic cell types appears to produce clusters of highly correlated cell types as demonstrated in Figure 5. After augmentation, the reported correlation increases (table 2) but the spillover matrix in Figure 6 seems to show that more cell types are being misclassified (a larger cluster of cell types in blue). The author should provide clarification.

2. The argument is made that performing the spillover clustering algorithm improves estimation of the relative order of cell type percentages at the expense of accuracy. However, it appears that the authors have only reported the Pearson correlation in Table 2 following clustering. To demonstrate that the rank order estimation has improved, it is necessary to include the Spearman correlation coefficient as well.

3. Augmentation of LM22 to include myeloma cell types is demonstrated to improve RMSE and the correlation coefficient when applied to two cell populations. The reported improvement in Pearson correlation after augmentation is minor though the reported RMSE is significantly reduced. By adding new cell types to the signature matrix that are supposedly at low abundance, it appears that the algorithm’s performance is not affected significantly. The author should include interpretation of these results that clarifies the role of RMSE and Pearson correlation coefficient as evaluation metrics for ADAPTS.

4. Adding new cell types to the signature matrix without adding new genes appears to significantly increase the matrix condition number (figure 1). This is despite the author’s assumption that these genes should not be informative for classifying other cell types. Further explanation as to why the condition number blows up so dramatically is required. Additional detail is required for the condition number minimization procedure, as well.

5. Using the top 100 differentially expressed genes between the average of the signature matrix cell types and the new cell types to seed the de novo matrix seems to be fine for adding a few cell types. However, as the number of cell types increases, shouldn’t this number also increase to ensure that the system is overdefined?

Minor:

1. Figures 2 and 3 require titles.

2. The color bars for all of the heatmap figures require annotation. The legend should clearly provide interpretation for the clusters of highly correlated cell types.

3. Do the heat maps represent variance or some other misclassification metric?

Reviewer #2: Tumor tissue microenvironment deconvolution is key to understand the interactions between cancer cells and its environment. Information derived from deconvolution can not only be applied to predict immune therapy patient outcome, but also reveal the mechanism of tumor cells in the heterogeneity environment. The accuracy of a cell type specific gene signature matrix is usually the core to most of the deconvolution methods. This paper proposed a novel augmented method ADAPTS to enforce the application of the signature matrix adaptive to different tissue or cancer types.

This study has the following novelties.

1. The authors focused on augmenting the signature matrix. Users can upload custom cell type specific expressed genes to the current signature matrix by providing cell type specific training data. This augment strategy can be applied to the most deconvolution methods in order to improve the performance.

2. The authors proposed a novel hierarchical deconvolution algorithm. ADAPTS implements a novel method for clustering cell types into groups that are hard to distinguish and then re-splitting those clusters using hierarchical deconvolution.

3. The authors provided a user friendly R package on the CRAN and github.

Nonetheless, further discussions of the following points are expected:

1. This method enables the addition of more custom cell type in the signature matrix. However, the authors need to address the co-linearity issue among the cell types. Adding more custom cell types (some of which may be highly similar to the existing cell types) would cause highly unstable results.

2. On page 2, line 43, the signature matrices are trained from samples of purified cell types. However, the deconvolution method aims to predicted immune cell infiltration of tumor. The immune cells have huge variations comparing to those in the purified cell line states. How could ADAPTS deal with the differences between the cancer suppressed immune cell and purified immune cell in the training data?

3. The authors showed that the augmented signature matrix based deconvolution has good prediction of WBM. The authors are expected to show how adding the new cell types in the current signature matrix will impact the prediction of the previously existing cell types.

4. In the signature matrix augmentation parts, the author selected additional genes that best differentiate each cell types. What if some important marker genes are expressed in two cell types?

5. On page 5, Eq 5 resulted in an initial spillover matrix, E_0. Here, each row of E_0 is a cell type and each column is a sample. Then, the average function A was applied to the matrix E_0 in order to get the matrix S_1. What is the dimension of the matrix S_1?

6. On page 5, Eq 7 used deconvolution function D again to calculate the new spillover matrix. If the second parameter of function D is transpose of E_0, what is the dimension of matrix E_1? It seems like each row of E_0 is a cell type and each column is a sample. The transpose of matrix E_0 would be a sample (row) by cell type (column) matrix. How the matrix E_1 can be in the same form of transpose of the matrix E_0?

7. On page 5, line 145, the algorithm converges in a clustered spillover matrix (Fig. 3). How to explain that T cell was clustered into two different clusters in the Fig. 3?

8. On page 5, line 127, Figure 2 should be Figure 3.

6. PLOS authors have the option to publish the peer review history of their article (what does this mean?). If published, this will include your full peer review and any attached files.

Reviewer #1: Yes: Xiling Shen

Reviewer #2: No

---

## [Author Response · Author response to Decision Letter 0]

25 Sep 2019

ADAPTS Responses to Reviewer Comments

We would like to thank the editor for their care in selecting reviewers for our manuscript. We appreciate the obvious time that the reviewers took to read our equations and provide thoughtful comments and have accordingly thanked them in the acknowledgments. 

We have replaced all of the figures with recently regenerated .pdf files so that we have high quality vector version of the figures. Due to the stochasticity introduced by the imputation in the augmentation algorithm, this has resulted in slightly different figures then in the previous submission, however the points conveyed by the figures have not changed. One figure (Figure 3) has changed because it was generated using an earlier version of the algorithm (i.e. an algorithm slightly different than that described in the publication). Again, the point of the figure has not changed. However, we apologize for any confusion caused by this sloppiness. 

We have also added a new overview figure to the introduction that we believe will help any reader more readily understand how ADAPTS works.

Reviewer #1: Cell type deconvolution is an innovative technique to achieve higher resolution from bulk transcriptomic data. In this manuscript, Danziger, et al., build on existing methods for profiling leukocyte infiltration of tumors by offering their package, ADAPTS, as a method to augment the reference signature matrix used to deconvolve bulk, heterogenous samples. The authors aimed to address two issues that plague implementation of deconvolution algorithms: the presence of stromal cells alongside immune cells that complicates analysis and the tendency for misclassification of closely related cell types, which they termed spillover. The ADAPTS package enables augmentation of existing signature matrices by including differentially expressed genes which define gene expression profiles for additional cell types and minimizes condition number, a measure of matrix stability. The ADAPTS package deals with spillover by iteratively performing deconvolution on highly correlated cell types then applies hierarchical deconvolution on a subset of the genes that define each cell type. The ADAPTS package aims to offer personalization of existing deconvolution techniques to an individual study, but the added value demonstrated in the included examples is questionable. Some clarification of the existing evidence is required.

Major:

1. Using the top 100 differentially expressed genes on average between the existing signature matrix cell types and the added pancreatic cell types appears to produce clusters of highly correlated cell types as demonstrated in Figure 5. After augmentation, the reported correlation increases (table 2) but the spillover matrix in Figure 6 seems to show that more cell types are being misclassified (a larger cluster of cell types in blue). The author should provide clarification.

The reviewer is quite correct that augmenting the signature matrix over the 'top 100' improves the correlation with actual cell counts, but also results in more cell types that are clustered together by the spillover matrix. In particular, the 'top 100' clusters 'unclassified…' [7.02 vs 0.40], 'co…' [4.36 vs 14.26], 'beta…' [7.11 vs 3.51], 'delta' [0.00 vs 0.88]. The augmented cell type clusters 'unclassified…' [6.11 vs 0.40], 'gamma'[8.51 vs 1.83], 'delta'[8.04 vs 0.88], 'co…'[8.41 vs 14.26], 'alpha'[8.34 vs 12.11], 'beta' [7.66 vs 3.51]. Within the subset of the clustered cells, the Pearson's correlation coefficient is 0.029 for the 'top 100' compared to 0.51 for the augmented. This could certainly be confusing so we offer the following clarification.

At the end of Section entitled 'Spillover to Convergence' we have added the following text:

Cell types co-clustered by this method are cell types that have similar deconvolution patterns from purified samples. Within cell type clusters a cell type will usually deconvolve most frequently as itself and less frequently as other cell types within the cluster (e.g. NK.cells.activated and NK.cells.resting in Fig \\ref{fig:02} and Fig \\ref{fig:03}). Thus clusters are not necessarily a block within which cellular proportions are inaccurate. Rather, co-clustered cell types would benefit from additional efforts to separate them, such as by finding genes that specifically differentiate those cell types as is done in hierarchical deconvolution. 

2. The argument is made that performing the spillover clustering algorithm improves estimation of the relative order of cell type percentages at the expense of accuracy. However, it appears that the authors have only reported the Pearson correlation in Table 2 following clustering. To demonstrate that the rank order estimation has improved, it is necessary to include the Spearman correlation coefficient as well.

We thank the reviewer for pointing out a section where we have mis-interpreted our own data.

In the section labeled 'Clustering Cell Types Improves Deconvolution Accuracy' we added the sentence.

Using Spearman's rank-order correlation, we saw only modest increases in the correlation coefficient: the Top100 Spearman's $\\rho$ increased from 0.50 to 0.51 and the Augmented Spearman's $\\rho$ increased 0.69 to 0.71. Thus the bulk of the improvement in Pearson's correlation came from effectively removing from consideration low abundance cell types that were estimated as 0\\% using the Top100 matrix or being over-estimated from the spillover of other cell types using the Augmented matrix.

3. Augmentation of LM22 to include myeloma cell types is demonstrated to improve RMSE and the correlation coefficient when applied to two cell populations. The reported improvement in Pearson correlation after augmentation is minor though the reported RMSE is significantly reduced. By adding new cell types to the signature matrix that are supposedly at low abundance, it appears that the algorithm’s performance is not affected significantly. The author should include interpretation of these results that clarifies the role of RMSE and Pearson correlation coefficient as evaluation metrics for ADAPTS.

The reviewer points to a clear problem with any error metric: it is a single measurement that describes the health of an entire system. Thus, it is fitting and appropriate that we explain how our error metrics apply to our problem and what their limitations are. Additionally, the other reviewer asked for us to show how changing the signature matrix changed the deconvolution of the non-tumor cell types, which are Table 2 and Table 3 in the revised manuscript. These tables allow a reader to go beyond the metric and to clearly see how the different cell types contribute the final error scores.

We have added the following subsection to Materials and Methods entitled: 'Evaluation Metrics'

To evaluate the efficacy of deconvolution, we use two common metrics: Pearson's correlation coefficient ($\\rho$) and root mean squared error (RMSE). Correlation represents the quality of the linear relationship between the predicted and actual cell type percentages. Root mean square error (RMSE) directly measures the error between the actual and predicted values for a cell type. A detailed discussion of these metrics and other alternatives are presented by Li and Wu\\cite{li_toast_2019}.

In \\nameref{ex:tumor}, correlation and RMSE evaluate the accuracy of predicting a single cell type (i.e. myeloma tumor) across multiple samples. In this kind of analysis, if a particular cell type has a high correlation, but also a high RMSE, that indicates that the estimates are systematically underestimating (or overestimating) the actual percentage of that cell type. In the case of underestimation, some percentage of that cell type is being misclassified as another cell type or cell types (and \\textit{vice versa} for overestimation). 

In \\nameref{ex:scRNAseq}, correlation and RMSE evaluate predictions for all cell types in a single sample. In this case, the aforementioned bias is not possible since both the predicted and actual cell percentages must add up to 100\\%.

Please also see the subsection 'Deconvolved Cell Types'

In the WBM samples (Table~\\ref{Tab:AllWBM}), adding 5 cell types without adding any additional genes results in little change to the average percentage of cell types excepting the reduction in 'Plasma.cells' percentage which appearently shift to 'PlasmaMemory'. Adding in additional genes to make MGSM27 greatly increases the estimates for the new stromal cell types (i.e. 'osteoclasts', 'osteoblasts', and adipocytes) as well as the tumor cell types largely at the expense of innate immune cells (e.g. Monocytes, Mast.cells, Neutrophils, \\textit{et al}.). This removes a bias where tumor cells were systematically under-represented (12.71\\% in LM22p5 versus 29.96\\% in MGSM27). The \\textit{de novo} MGSM27 results are similar to MGSM27, expect for an increase in certain types of T cells and a shifting in innate populations.

4. Adding new cell types to the signature matrix without adding new genes appears to significantly increase the matrix condition number (figure 1). This is despite the author’s assumption that these genes should not be informative for classifying other cell types. Further explanation as to why the condition number blows up so dramatically is required. Additional detail is required for the condition number minimization procedure, as well.

We thank the reviewer for giving us the expand on Algorithm #1 and the condition number. As the condition number is an important tool for limiting collinearity between cell type signatures in the signature matrix it is an important component of ADAPTS and deserve additional attention.

After introducing Algorithm #1, we have added a section entitled 'Condition Number Minimization and Smoothing' that states:

The condition number ($CN$) is calculated by the $\\kappa()$ function. In linear regression, $CN$ is a metric that increases with multicollinearity; in this case, how well can the signature of cell types be linearly predicted from the other cell types in the signature matrix. To illustrate this, it is helpful restate Eq \\ref{eq:01} using a signature matrix $S$ that has the same number of genes as the data to deconvolve $X$ and use the trivial deconvolution function: $D(S,X) = S^{-1}X$. This recasts Eq~\\ref{eq:01} deconvolution as the matrix decomposition problem presented in Eq~\\ref{eq:mat01}\\cite{gaujoux_cellmix:_2013}.

\\begin{equation}\\label{eq:mat01}

X \\approx SE

\\end{equation}

When the problem is stated in this manner, the $CN$ approximately bounds the inaccuracy $E$, the estimate of cellular composition\\cite{belsley_regression_2005}, and it remains meaningful if the system becomes underdetermined \\cite{degot_condition_2001}. 

By definition, the condition number will increase as the system become more multicollinear. If a signature matrix is augmented with new cell types that express the genes in the matrix in a pattern similar to other cell types in the matrix then the condition number would be expected to dramatically increase compared to the un-augmented matrix. This indicates a signature matrix lacking genes informative for differentiating multicollinear gene signatures. As Algorithm~\\ref{alg:aug} iteratively adds the top gene for each cell type, the condition number would be expected to decrease as the new genes are selected to differentiate that cell type from all other cell types. 

In practice, Algorithm~\\ref{alg:aug} sometimes results in clearly unstable minima, where the $CN$ decreases dramatically for one iteration only to increase dramatically the next. To avoid this instability, \\textbf{ADAPTS} smooths the $CN$ curve using Tukey's Running Median Smoothing (3RS3R)\\cite{beyer_tukey_1981}. Often, the $CN$s will decrease in very small increments for many iterations before beginning to rise, resulting thousands of genes in the signature matrix. A signature matrix ($S$) with more genes ($|g_S|$) than samples in the training data ($\\sum_{j \\in J}|J_j|$) essentially represents the solution to an underdetermined system that is likely to be overfit to the training data, resulting in reduced deconvolution accuracy one new samples. To mitigate this, a optional tolerance may be applied to find the minimum number of genes that has a $CN$ within some \\% of the true minimum. By default, \\textbf{ADAPTS} uses a 1\\% tolerance.

5. Using the top 100 differentially expressed genes between the average of the signature matrix cell types and the new cell types to seed the de novo matrix seems to be fine for adding a few cell types. However, as the number of cell types increases, shouldn’t this number also increase to ensure that the system is overdefined?

We address this question in our answer to question #4. Jerome Degot demonstrated in "A condition number theorem for underdetermined polynomial systems" Mathematic of Computation (2000) that condition numbers are meaningful for underdetermined systems. Practically speaking, the R kappa function will not crash if it applied to an underdetermined system. 

Minor:

1. Figures 2 and 3 require titles.

Titles have been added.

2. The color bars for all of the heatmap figures require annotation. The legend should clearly provide interpretation for the clusters of highly correlated cell types.

Legends have been revised.

3. Do the heat maps represent variance or some other misclassification metric?

Heatmaps represent percentage of cells classified as. For example, in Figure 2 if the row is 'B.cell.memory', the column is 'Plasma.cells', and the color is light blue then purified 'B.cell.memory' deconvolve as containing (on average) ~18% 'Plasma.cells'. 

The captions for Figure 2, 3, 5, and 6 have been revised to make this clearer.

Reviewer #2: Tumor tissue microenvironment deconvolution is key to understand the interactions between cancer cells and its environment. Information derived from deconvolution can not only be applied to predict immune therapy patient outcome, but also reveal the mechanism of tumor cells in the heterogeneity environment. The accuracy of a cell type specific gene signature matrix is usually the core to most of the deconvolution methods. This paper proposed a novel augmented method ADAPTS to enforce the application of the signature matrix adaptive to different tissue or cancer types.

This study has the following novelties.

1. The authors focused on augmenting the signature matrix. Users can upload custom cell type specific expressed genes to the current signature matrix by providing cell type specific training data. This augment strategy can be applied to the most deconvolution methods in order to improve the performance.

2. The authors proposed a novel hierarchical deconvolution algorithm. ADAPTS implements a novel method for clustering cell types into groups that are hard to distinguish and then re-splitting those clusters using hierarchical deconvolution.

3. The authors provided a user friendly R package on the CRAN and github.

Nonetheless, further discussions of the following points are expected:

1. This method enables the addition of more custom cell type in the signature matrix. However, the authors need to address the co-linearity issue among the cell types. Adding more custom cell types (some of which may be highly similar to the existing cell types) would cause highly unstable results.

We agree with the reviewer that co-linearity between cell types could certainly cause instability in the deconvolution.

One might imagine different algorithms that would potentially improve this situation. For instance, cell types that are highly co-linear (as detected by the spill-over clustering or other method) might be pooled together into a common cell type for purposes of determining differentially expressed genes. This could then be used for the initial deconvolution to estimate the percentage of cells in each cluster, then those percentages could re-partitioned based on the hierarchical clustering. We hope by having made ADAPTS both modular and on GitHub, users will be able to easily test out variations on the algorithm using ADAPTS and easily suggest improvements to be added to the core package.

We have added the Discussion section: Co-linear Cell Types that has the following text

In \\nameref{ex:tumor}, tumor cells are represented by four cell types with very similar gene expression patterns and this apparently creates problems, especially in \\textit{de novo} signature matrix construction. Eq \\ref{eq:rankScore} scores rank genes for each cell type to be iteratively added to the signature matrix by Algorithm \\ref{alg:aug}. When Eq \\ref{eq:rankScore} scores a set of cell types where a subset of the cell types have similar gene expression, it may underscore genes that are helpful for differentiating that subset from all other cell types. 

One might imagine algorithms that would potentially improve this situation by either increasing the score of genes common to subsets of cell types, or make certain that these genes appear in the seed matrix. For instance, cell types that are highly co-linear (as detected by the spill-over clustering or other method) might be pooled together into a common cell type for purposes of determining gene scores with the \\textbf{ADAPTS} function \\textbf{rankByT}. While high variance genes may already be present in the seed matrix, perhaps the gene scores should be up-weighted for genes that have a globally high variance; this would improve the representation of genes that are high in all tumor cell types (for example), but low in all others. Or perhaps genes might be scored in \\textbf{rankByT} by running pair-wise comparisons between all cell types, and the genes score for a particular cell type might be its top ratio against \\textit{any} other cell type rather than \\textit{all} other cell types. 

Ultimately, solving this problem is beyond the scope of this publication, however the \\textbf{ADAPTS} framework can help to test potential solutions which may then be shared via \\textbf{GitHub} and potentially included in the official \\textbf{CRAN} package.

2. On page 2, line 43, the signature matrices are trained from samples of purified cell types. However, the deconvolution method aims to predicted immune cell infiltration of tumor. The immune cells have huge variations comparing to those in the purified cell line states. How could ADAPTS deal with the differences between the cancer suppressed immune cell and purified immune cell in the training data?

The reviewer has pointed out one of the difficulties inherent to the deconvolution field. Templates (e.g. LM22) are based on cells purified from peripheral blood monocytes and those immune cells are expected to be quite different from the same cell type present in a tumor microenvironment. Fundamentally, if we do not know what genes are differentially expressed when an immune cell is in a tumor, we can not know what genes should be in the signature matrix. 

Fortunately, ADAPTS is designed for this ecosystem. As gene expression data for tumor-specific cell types become available, it would be straight-forward to remove the PBMC version of a cell-type and then add in the tumor-specific cell-type using ADAPTS. A new signature matrix with the new data could be constructed ab initio. For example, if a tumor sample is subjected to a scRNAseq experiment, then individual cells will be assigned to clusters which roughly correlate to cell-types. These clusters of cells can then be used in ADAPTS to augment an existing signature matrix or to construct a new matrix ab initio.

See the following in the Discussion Subsection 'Cancer Suppressed Immune Cells' 

Most gene expression data generated from purified immune cells, and thus most publicly available signature matrices, are generated from peripheral blood. However, immune and stromal cells in a tumor microenvironment are expected to differ in gene expression from those purified from non-tumor sources \\cite{gajewski_innate_2013}. If gene expression of cells purified from the relevant tumor microenvironment is available, it would be straight-forward to use that data in \\textbf{ADAPTS}. Either the tumor-associated cell types could be used to augment an existing signature matrix, or the relevant cell type (and the top genes for that cell type) could be removed from the signature matrix before \\textbf{ADAPTS} augments the cell type back in to the signature matrix using the new data. 

If a tumor microenvironment sample has been subjected to a scRNAseq experiment, then this has the potential to build a more accurate signature matrix than one from any purified cell type data. Individual cells will be assigned to clusters which roughly correlate with purified cell types. Theoretically, these cell type clusters would represent the immune, stromal, and tumor cell gene expression in the microenvironment and enumerate exactly the cell types present in the sample. See \\nameref{ex:scRNAseq} for an example of how to construct a signature matrix \\textit{de novo} from single cell data using \\textbf{ADAPTS}. 

3. The authors showed that the augmented signature matrix based deconvolution has good prediction of WBM. The authors are expected to show how adding the new cell types in the current signature matrix will impact the prediction of the previously existing cell types.

We thank the author for pointing out this oversight. We have added a new Table 2 and Table 3 which show the average for each deconvolved cell-types for the CD138+ and WBM samples. These cell types are discussed in the new subsection Deconvolved Cell-Types. 

Enumerating the cell types detected by deconvolution in the CD$138^{+}$ and WBM samples by the four signature matrices illustrates the changes brought about by augmenting the signature matrix. Table~\\ref{Tab:AllCD138p} shows the mean percentage of each cell type across the 423 CD$138^{+}$ samples and Table~\\ref{Tab:AllWBM} shows the same for the 440 WBM samples.

In the CD$138^{+}$ samples (Table~\\ref{Tab:AllCD138p}), adding 5 cell types without adding any additional genes has the immediate benefit of greatly reducing the percentage of cells classed as immune cells that should not be in the sample (e.g. 'T.cells.follicular.helper', 'Monocytes'). However, this is balanced out by adding percentages of new cell types that should not be in the samples (e.g. 'adipocyte') and deconvolving only a relatively small percentage (4.62\\%) as the most obvious tumor type: 'MM.plasma.cell'. This is consistent with the hypothesis that the signature matrix lacks the correct genes to precisely identify these cell types. Adding in additional genes to make MGSM27 moves cancer types 'MM.plasma.cell', 'Plasma.cells', and 'PlasmaMemory' cells to the most frequent cell types and reduces all other cell types to very low estimates. The \\textit{de novo} MGSM27 shows an increase in many cell types that should not be in the samples. One explanation for this might be that the genes common to 'B.cells.memory', 'Plasma.cells', 'PlasmaMemory', and 'MM.plasma.cell' are scored inappropriately lowly by Eq \\ref{eq:rankScore}.

In the WBM samples (Table~\\ref{Tab:AllWBM}), adding 5 cell types without adding any additional genes results in little change to the average percentage of cell types excepting the reduction in 'Plasma.cells' percentage which appearently shift to 'PlasmaMemory'. Adding in additional genes to make MGSM27 greatly increases the estimates for the new stromal cell types (i.e. 'osteoclasts', 'osteoblasts', and adipocytes) as well as the tumor cell types largely at the expense of innate immune cells (e.g. Monocytes, Mast.cells, Neutrophils, \\textit{et al}.). This removes a bias where tumor cells were systematically under-represented (12.71\\% in LM22p5 versus 29.96\\% in MGSM27). The \\textit{de novo} MGSM27 results are similar to MGSM27, expect for an increase in certain types of T cells and a shifting in innate populations. 

4. In the signature matrix augmentation parts, the author selected additional genes that best differentiate each cell types. What if some important marker genes are expressed in two cell types?

Please see the answer to comment 1, which we believe addresses this concern.

5. On page 5, Eq 5 resulted in an initial spillover matrix, E_0. Here, each row of E_0 is a cell type and each column is a sample. Then, the average function A was applied to the matrix E_0 in order to get the matrix S_1. What is the dimension of the matrix S_1?

We apologize for creating confusion by sloppily defining A(P). Suppose P has |C| cell types and each cell type has |P^i| samples where i = 1:|C|. A(P) results in a matrix with |C| columns and A(P^i) results in a matrix with 1 column. 

In this case, the dimensions of S_1 has rows equal to the number of cell types + 1 for the additional 'others' column, and columns equal to the number of cell types . 

We have removed the definition of A(P^1) from around line 58 so that the newly expanded definition of A(P) after Algorithm #1 (around line 90) is more salient.

$A(P)$ returns the mean expression for each gene in each cell type, producing a matrix such as is shown on the right side of Eq~\\ref{eq:02}. If $P$ has $|C|$ cell types, $|G|$ genes, and each cell type has some number of samples, $|P^i|$ where $i = 1:|C|$, then $A(P)$ would result in a matrix with $|C|$ columns and $|G|$ rows. When $A(P)$ is called on a matrix with one cell type, $P^i$, then $A(P^i)$ results in a matrix with one column and $|G|$. 

The text after Equation 5 has been amended:

Thus $E^0$ would have $|C|$ rows representing each cell type in $S^0$ and one column for each of the $|P^0|$ samples. Applying $A(E^0)$ to average the cell type estimates $E$ across purified samples makes the spillover matrix resemble a signature matrix, leading to Eq \\ref{eq:05}.

The text after Equation 6 has been amended:

This new spillover based deconvolution matrix $S^1$ has $|C|$ rows with the average percentage that each of the $|C|$ purified cell types has deconvolved into. $S^1$ can be used to re-deconvolve the initial spillover matrix, $E^0$, effectively 'sharpening' the deconvolution matrix image as shown in Eq \\ref{eq:06}. 

6. On page 5, Eq 7 used deconvolution function D again to calculate the new spillover matrix. If the second parameter of function D is transpose of E_0, what is the dimension of matrix E_1? It seems like each row of E_0 is a cell type and each column is a sample. The transpose of matrix E_0 would be a sample (row) by cell type (column) matrix. How the matrix E_1 can be in the same form of transpose of the matrix E_0?

We thank the review for giving us the opportunity to add additional clarity to an unnecessarily confusion section, including removing an embarrassing error. 

Equation 7 should be E^1 = D(S^1, E^0)

Thus,

Thus $E^1$ will have $|C|$ rows taken from the $|C|$ columns in $S^1$ and $|P^0|$ columns taken from the columns in $E^0$. Once these values are calculated, the following pseudocode (Algorithm \\ref{alg:02}) shows how \\textbf{ADAPTS} iteratively applies spillover re-deconvolution to cluster cell types likely to be confused by deconvolution.

And within Algorithm 2, line 5 has been amended to be E^i = D(S^i, E^(i-1))

7. On page 5, line 145, the algorithm converges in a clustered spillover matrix (Fig. 3). How to explain that T cell was clustered into two different clusters in the Fig. 3?

The reviewer points to a fundamental difficulty in using gene expression analysis to analyze immune cells. Immune cells are classified primarily by surface markers, for example CD3 for T-cells with additional markers specifying subsets of T-cells. The cells are not defined based on the genes that they actively expressing. Thus, one likely explanation for having three different clusters of T-cells is that each type of T-cell is expressing different genes at different levels. For the subset of the genes in the signature matrix the co-clustered T-cells have similar expression patterns, but each of the three clusters has different expression patterns.

We have revised the section to include the following sentence.

Similarly, T cells are broken into three blocks, implying that were T cells classified by gene expression rather than surface markers, the broad T cell families (i.e. $CD4^+$ and $CD8^+$) might be differently defined.

8. On page 5, line 127, Figure 2 should be Figure 3.

We thank the reviewer for pointing out this error.

Line 127 has been amended to say 'Figure 3'

---

## [Decision Letter · Decision Letter 1]

21 Oct 2019

ADAPTS: Automated Deconvolution Augmentation of Profiles for Tissue Specific cells

PONE-D-19-19603R1

Dear Dr. Danziger,

We are pleased to inform you that your manuscript has been judged scientifically suitable for publication and will be formally accepted for publication once it complies with all outstanding technical requirements. **There is a minor comment from reviewer #2 regarding two tables and the authors are strongly suggested to make changes so that the results would be presented in a more meaningful way.**

With kind regards,

Sha Cao, Ph.D

Academic Editor

PLOS ONE

Additional Editor Comments (optional):

Reviewers' comments:

Reviewer's Responses to Questions

**Comments to the Author**

1. If the authors have adequately addressed your comments raised in a previous round of review and you feel that this manuscript is now acceptable for publication, you may indicate that here to bypass the “Comments to the Author” section, enter your conflict of interest statement in the “Confidential to Editor” section, and submit your "Accept" recommendation.

Reviewer #2: All comments have been addressed

2. Is the manuscript technically sound, and do the data support the conclusions?

Reviewer #2: Yes

3. Has the statistical analysis been performed appropriately and rigorously? 

Reviewer #2: Yes

4. Have the authors made all data underlying the findings in their manuscript fully available?

Reviewer #2: Yes

5. Is the manuscript presented in an intelligible fashion and written in standard English?

Reviewer #2: Yes

6. Review Comments to the Author

Reviewer #2: The author addressed all my concern and answered them in great detail.

For the question 2 of review 2, the author mentioned that ADAPTS has the capability that handle the inaccurate signature matrix by adding new cell type signature and removing old biased cell type signature. This gives the promising direction to bring the power of single cell sequencing with high resolution in the specific cancer tissue environment.

Minor:

For the question 3 of review 2, the author added Table 2 and Table 3 to illustrate the posterior result of all existing cell types after adding custom cell types and adding more genes in the current signature matrix. However, these two table show the average deconvolution percentages of all cell types for two experiment datasets. This reconstruction values just show the trend of predicted cell type proportion which cannot reflect the relationship with the ground truth (actual cell type percentages). It is necessary to include the correlation as well. Or the comparison with the known percentage is required to add.

7. PLOS authors have the option to publish the peer review history of their article (what does this mean?). If published, this will include your full peer review and any attached files.

Reviewer #2: Yes: Wennan Chang

---

## [Editor Report · Acceptance letter]

7 Nov 2019

PONE-D-19-19603R1 

ADAPTS: Automated Deconvolution Augmentation of Profiles for Tissue Specific cells 

Dear Dr. Danziger:

I am pleased to inform you that your manuscript has been deemed suitable for publication in PLOS ONE. Congratulations! Your manuscript is now with our production department. 

With kind regards,

on behalf of

Dr. Sha Cao 

Academic Editor

PLOS ONE